# VCNet and Functional Targeted Regularization For Learning Causal Effects of Continuous Treatments

**Lizhen Nie**[*1], **Mao Ye**[*2], **Qiang Liu**[2], **Dan Nicolae**[1]
[1]Department of Statistics, The University of Chicago
[2]Department of Computer Science, University of Texas at Austin
`lizhen@statistics.uchicago.edu, my21@cs.utexas.edu,`
`lqiang@cs.utexas.edu, nicolae@statistics.uchicago.edu`

## ABSTRACT

Motivated by the rising abundance of observational data with continuous treatments, we investigate the problem of estimating the average dose-response curve (ADRF). Available parametric methods are limited in their model space, and previous attempts in leveraging neural network to enhance model expressiveness relied on partitioning continuous treatment into blocks and using separate heads for each block; this however produces in practice discontinuous ADRFs. Therefore, the question of how to adapt the structure and training of neural network to estimate ADRFs remains open. This paper makes two important contributions. First, we propose a novel varying coefficient neural network (VCNet) that improves model expressiveness while preserving continuity of the estimated ADRF. Second, to improve finite sample performance, we generalize targeted regularization to obtain a doubly robust estimator of the whole ADRF curve.

## 1 INTRODUCTION

Continuous treatments arise in many fields, including healthcare, public policy, and economics. With the widespread accumulation of observational data, estimating the average dose-response function (ADRF) while correcting for confounders has become an important problem (Hirano & Imbens, 2004; Imai & Van Dyk, 2004; Kennedy et al., 2017; Fong et al., 2018).

Recently, papers in causal inference (Johansson et al., 2016; Alaa & van der Schaar, 2017; Shalit et al., 2017; Schwab et al., 2019; Farrell et al., 2018; Shi et al., 2019) have utilized feed forward neural network for modeling. The success of using neural network model lies in the fact that neural networks, unlike traditional parametric models, are very flexible in modeling the complex causal relationship as shown by the universal approximation theorem (Csáji et al., 2001). Also, unlike traditional non-parametric models, neural network has been shown to be powerful when dealing with high-dimensional input (i.e., Masci et al. (2011); Johansson et al. (2016)), which implies its potential for dealing with high-dimensional confounders.

A successful application of neural network to causal inference requires a specially designed network structure that distinguishes the treatment variable from other covariates, since otherwise the treatment information might be lost in the high dimensional latent representation (Shalit et al., 2017). However, most of the existing network structures are designed for binary treatments and are difficult to generalize to treatments taking value in continuum. For example, Shalit et al. (2017); Louizos et al. (2017); Schwab et al. (2019); Shi et al. (2019) used separate prediction heads for the two treatment options and this structure is not directly applicable for continuous treatments as there is an infinite number of treatment levels. To deal with a continuous treatment, recent work (Schwab et al., 2019) proposed a modification called DRNet. DRNet partitions a continuous treatment into blocks and for each block, trains a separate head, in which the treatment is concatenated into each hidden layer (see Figure 2). Despite the improvements made by the building block of DRNet, this structure does not

---

*Equal contribution and corresponding authors.

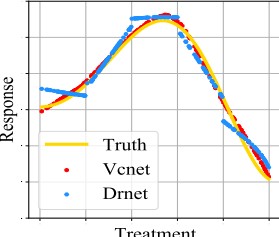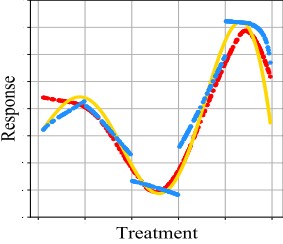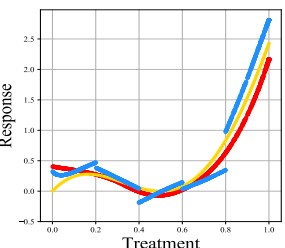

Figure 1: Estimated ADRF on testing set from a typical run of VCNet and DRNet. From left to right panels are results on simulation, IHDP and News dataset. Both VCNet and DRNet are well optimized. Blue points denote DRNet estimation and red points VCNet. The truth is shown in yellow solid line.

take the continuity of ADRF (Prichard & Gillam, 1971; Schneider et al., 1993; Threlfall & English, 1999) into account, and it produces discontinuous ADRF estimators in practice (see Figure 1).

We propose a new network building block that is able to strengthen the influence of treatment but also preserve the continuity of ADRF. Under binary treatment, previous neural network models for treatment effect estimation use separate prediction heads to model the mapping from covariates to (expected) outcome under different treatment levels. When it comes to the continuous treatment case, by the continuity of ADRF, this mapping should change continuously with respect to the treatment. To achieve this, motivated by the varying coefficient model (Hastie & Tibshirani (1993), Fan et al. (1999), Chiang et al. (2001)), one can allow the weights of the prediction head to be continuous functions of the treatment. This serves as the first contribution here, called Varying Coefficient Network (VCNet). In VCNet, once the activation function is continuous, the mapping defined by the network automatically produces continuous ADRF estimators as shown in Figure 1 but also prevents the treatment information from being lost in its high dimensional latent representation.

The second contribution of this paper is to generalize targeted regularization (Shi et al., 2019) to obtain a doubly robust estimator of the whole ADRF curve, which improves finite sample performance. Targeted regularization was previously used for estimating a scalar quantity (Shi et al., 2019) and it associates an extra perturbation parameter to the scalar quantity of interest. While adapting it to a finite-dimensional vector is not difficult, generalization to a curve is far less straightforward. Difficulties arise from the fact that ADRFs cannot be regularized at each treatment level independently because the number of possible levels is infinite and, thus, the model complexity cannot be controlled with the introduction of infinite extra perturbations parameters. Utilizing the continuity (and smoothness) of ADRF (Schneider et al., 1993; Threlfall & English, 1999), we introduce smoothing to control model complexity. Its model size increases in a specific manner to balance model complexity and regularization strength. Moreover, the original targeted regularization in Shi et al. (2019) is not guaranteed to obtain a doubly robust estimator. By allowing regularization strength to depend on sample size, we obtain a consistent and doubly robust estimator under mild assumptions. Noticing the connection between targeted regularization and TMLE (Van Der Laan & Rubin, 2006), a by-product of this work is that we give the first (to the best of our knowledge) generalization of TMLE to estimating a function.

We do experiments on both synthetic and semi-synthetic datasets, finding that VCNet and targeted regularization boost performance independently. Using them jointly consistently achieves state-of-the-art performance.

**Notation** We denote the Dirac delta function by $\delta(\cdot)$. We use $\mathbb{E}$ to denote expectation, $\mathbb{P}$ to denote population probability measure and we write $\mathbb{P}(f) = \int f(z)d\mathbb{P}(z)$. Similarly, we denote $\mathbb{P}_n$ as the empirical measure and we write $\mathbb{P}_n(f) = \int f(z)d\mathbb{P}_n(z)$. We denote $\lceil n \rceil$ as the least integer greater than or equal to $n$, and we denote $\lfloor n \rfloor$ as the greatest integer less than or equal to $n$. We use $\tau$ to denote Rademacher random variables. We denote Rademacher complexity of a function class $\mathcal{F} : \mathcal{X} \to \mathbb{R}$ as $\mathrm{Rad}_n(\mathcal{F}) = \mathbb{E}\left(\sup_{f \in \mathcal{F}} \left| \frac{1}{n} \sum_{i=1}^{n} \tau_i f(X_i) \right|\right)$. Given two functions $f_1, f_2 : \mathcal{X} \to \mathbb{R}$, we define $\|f_1 - f_2\|_\infty = \sup_{x \in \mathcal{X}} |f_1(x) - f_2(x)|$ and $\|f_1 - f_2\|_{L^2} = \left(\int_{x \in \mathcal{X}} (f_1(x) - f_2(x))^2 \, dx\right)^{1/2}$. For a function class $\mathcal{F}$, we define $\|\mathcal{F}\|_\infty = \sup_{f \in \mathcal{F}} \|f\|_\infty$. We denote stochastic boundedness with $O_p$

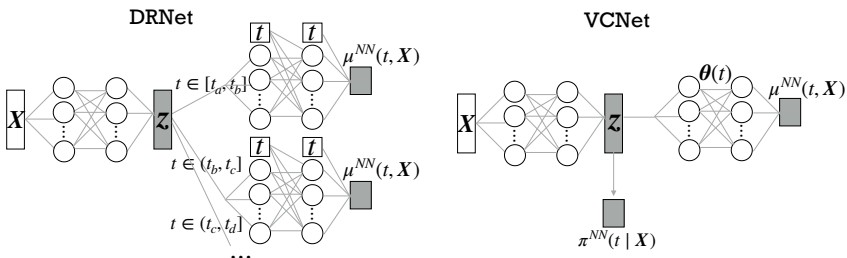

Figure 2: Comparison of network structure between DRNet and VCNet.

and convergence in probability with $o_p$. Given two random variable $X_1$ and $X_2$, $X_1 \perp X_2$ denotes $X_1$ and $X_2$ are independent. We use $a_n \asymp b_n$ to denote that both $a_n/b_n$ and $b_n/a_n$ are bounded.

## 2 PROBLEM STATEMENT AND BACKGROUND

Suppose we observe an i.i.d sample $\{(y_i, \boldsymbol{x}_i, t_i)\}_{i=1}^n$ where $(y_i, \boldsymbol{x}_i, t_i)$ is a realization of random vector $(Y, \boldsymbol{X}, T)$ with support $(\mathcal{Y} \times \mathcal{X} \times \mathcal{T})$. Here $\boldsymbol{X}$ is a vector of covariates, $T$ is a continuous treatment, and $Y$ is the outcome. Without loss of generality, we assume $\mathcal{T} = [0, 1]$. We want to estimate the Average Dose Response Function (ADRF)

$$\psi(t) := \mathbb{E}(Y \mid \mathrm{do}(T = t)),$$

which is the potential expected outcome that would have been observed under treatment level $t$. Suppose the conditional density of $T$ given $\boldsymbol{X}$ is $\pi(T \mid \boldsymbol{X})$. Throughout this paper, we make the following assumptions:

**Assumption 1.** *(a) There exists some constant $c > 0$ such that $\pi(t \mid \boldsymbol{x}) \geq c$ for all $\boldsymbol{x} \in \mathcal{X}$ and $t \in \mathcal{T}$. (b) The measured covariate $\boldsymbol{X}$ blocks all backdoor paths of the treatment and outcome.*

**Remark 1.** *Assumption (a) implies that treatment is assigned in a way that every subject has some chance of receiving every treatment level regardless of its covariates, which is a standard assumption to establish doubly robust estimators. Assumption (b) implies that the casual effect is identifiable, i.e., can be estimated using observational data.*

## 3 VCNET: VARYING COEFFICIENT NETWORK STRUCTURE

Under Assumption 1, we have

$$\psi(t) = \mathbb{E}\left[\mathbb{E}(Y \mid \boldsymbol{X}, T = t)\right].$$

Thus, a naive estimator for $\psi$ is to obtain an estimator $\hat{\mu}$ of $\mu$, and use $\hat{\psi}(t) = \frac{1}{n} \sum_{i=1}^n \hat{\mu}(t, \boldsymbol{x}_i)$. Here $\mu(t, \boldsymbol{x}) := \mathbb{E}(Y \mid \boldsymbol{X} = \boldsymbol{x}, T = t)$ and $\hat{\mu}$ is its estimator. Following Shi et al. (2019), we utilize the sufficiency of the generalized propensity score $\pi(t \mid \boldsymbol{X})$ for estimating $\psi$ (Hirano & Imbens, 2004):

$$\psi(t) = \mathbb{E}\left[\mathbb{E}(Y \mid \pi(t \mid \boldsymbol{X}), T = t)\right].$$

It indicates that learning $\pi(t \mid \boldsymbol{X})$ helps the removal of noise and distillation of useful information in $\boldsymbol{X}$ for estimating $\psi$. Similar to Shi et al. (2019), we add a separate head for estimating $\pi(t \mid \boldsymbol{X})$, and use the feature $\boldsymbol{z}$ extracted by it for downstream estimation of $\mu(t, \boldsymbol{x})$ (see Figure 2). Our contribution here is to propose the varying coefficient structure of the prediction head for $\mu(t, \boldsymbol{x})$, which addresses difficulties confronting continuous treatment as discussed in the following paragraph.

### 3.1 THE VARYING COEFFICIENT PREDICTION HEAD

Our aim is to predict $\mu(t, \boldsymbol{x}) = \mathbb{E}(Y \mid T = t, \boldsymbol{X} = \boldsymbol{x})$. A naive method is to train a neural network which takes $(t, \boldsymbol{x})$ as input in the first layer and outputs $\mu(t, \boldsymbol{x})$ in the last layer. However, the role of treatment $t$ is different from that of $\boldsymbol{x}$ and the influence of $t$ might be lost in the high-dimensional hidden features (Shalit et al., 2017). Aware of this problem, previous work (Schwab et al., 2019)

divides the range of treatment into blocks, and then use separate prediction heads for each block (see Figure 2). To further strengthen the influence of treatment $t$, Schwab et al. (2019) appends $t$ to each hidden layer. One problem of this structure is that it destroys the continuity of $\mu$ by using different prediction heads for each block of treatment levels. In practice, DRNet indeed produces discontinuous curve (see Figure 1).

In order to simultaneously emphasize the influence of treatment while preserving the continuity of ADRF, we propose a varying coefficient neural network (VCNet). In VCNet, the prediction head for $\mu$ is defined as

$$\mu^{\mathrm{NN}}(t, \boldsymbol{x}) = f_{\boldsymbol{\theta}(t)}(\boldsymbol{z}),$$

where input $\boldsymbol{z}$ is the feature extracted by the conditional density estimator, and $f_{\boldsymbol{\theta}(t)}$ is a (deep) neural network with parameter $\boldsymbol{\theta}(t)$ instead of a fixed $\boldsymbol{\theta}$. It means that the nonlinear function defined by the neural network depends on the varying treatment level $t$, and thus we call this structure the varying coefficient structure (Hastie & Tibshirani, 1993; Fan et al., 1999; Chiang et al., 2001). For example, if $f$ is an one-hidden-layer ReLU network, we have $f_{\boldsymbol{\theta}(t)}(\boldsymbol{z}) = \sum_{i=1}^{D} a_i(t) \mathrm{ReLU}(\boldsymbol{b}_i(t)^\top \boldsymbol{z})$, where $a_i(t)$ and $\boldsymbol{b}_i(t)$ are weights of the neural network, and $\boldsymbol{\theta}(t) = [(a_1(t), \boldsymbol{b}_1(t)), \cdots, (a_D(t), \boldsymbol{b}_D(t))]^\top$. Here we use splines to model $\boldsymbol{\theta}(t)$. Suppose that $\boldsymbol{\theta}(t) = [\theta_1(t), ..., \theta_{d_{\boldsymbol{\theta}}}(t)]^\top \in \mathbb{R}^{d_{\boldsymbol{\theta}(t)}}$, where $d_{\boldsymbol{\theta}(t)}$ is the dimension of $\boldsymbol{\theta}(t)$. We have

$$\theta_i(t) = \sum_{l=1}^{L} a_{i,\ell} \varphi_\ell^{\mathrm{NN}}(t),$$

where $\{\varphi_l^{\mathrm{NN}}\}_{\ell=1}^{L}$ are the spline basis and $a_{i,\ell}$'s are the coefficients. Thus, we have

$$\boldsymbol{\theta}(t) = \mathbf{A}\boldsymbol{\Phi}(t) \quad \text{where} \quad \mathbf{A} = \begin{bmatrix} a_{1,1} & \cdots & a_{1,L} \\ \vdots & \ddots & \vdots \\ a_{d_{\boldsymbol{\theta}},1} & \cdots & a_{d_{\boldsymbol{\theta}},L} \end{bmatrix} \quad \text{and} \quad \boldsymbol{\Phi}(t) = \left[\varphi_1^{\mathrm{NN}}(t), \cdots, \varphi_L^{\mathrm{NN}}(t)\right]^\top.$$

It is worth mentioning that by choosing spline basis of the form $\mathbb{I}(t_0 \leq t < t_1)$ with different $t_0, t_1$, we recover the structure in Schwab et al. (2019), which has a separate prediction head for each block. It indicates that DRNet can also be viewed as a special case of VCNet (under a suboptimal choice of basis functions).

In VCNet, the influence of treatment effect $t$ on the outcome directly enters through parameters $\boldsymbol{\theta}(t)$ of the neural network, which distinguishes treatment from the other covariates and avoids the treatment information from being lost. Under typical choices of spline basis such as B-spline, once the activation function is continuous, VCNet will automatically produce continuous ADRF estimators.

## 3.2 Conditional Density Estimator

Recall that the input feature $\boldsymbol{z}$ to $\mu^{\mathrm{NN}}$ is extracted by the conditional density estimator for $\pi(t \mid \boldsymbol{x})$. Here we propose a simple network to estimate $\pi$, which is a direct generalization of the conditional probability estimating head in Shi et al. (2019). Notice that $t \in [0, 1]$ and the conditional density $\pi(t \mid \boldsymbol{x})$ is continuous with respect to treatment $t$ for any given $\boldsymbol{x}$. A continuous function can be effectively approximated by piecewise linear functions. Thus, we divide $[0, 1]$ equally into $B$ grids, estimate the conditional density $\pi(\cdot \mid \boldsymbol{x})$ on the $(B + 1)$ grid points, and the conditional density for other $t$'s are calculated via linear interpolation. To be more specific, we define the network $\pi_{\mathrm{grid}}^{\mathrm{NN}}$ as

$$\pi_{\mathrm{grid}}^{\mathrm{NN}}(\boldsymbol{x}) = \mathrm{softmax}(\boldsymbol{\omega}_2 \boldsymbol{z}) \in \mathbb{R}^{B+1}, \quad \text{where} \quad \boldsymbol{z} = f_{\boldsymbol{\omega}_1}(\boldsymbol{x}).$$

Here $\boldsymbol{z} \in \mathbb{R}^h$ is the hidden feature extracted by the network, $\boldsymbol{\omega}_1$ is the parameter for the nonlinear mapping $f_{\boldsymbol{\omega}_1}$, $\boldsymbol{\omega}_2 \in \mathbb{R}^{(B+1) \times h}$, $\pi_{\mathrm{grid}}^{\mathrm{NN}}(\boldsymbol{x}) = [\pi_{\mathrm{grid}}^{0,\mathrm{NN}}(\boldsymbol{x}), ...., \pi_{\mathrm{grid}}^{B,\mathrm{NN}}(\boldsymbol{x})]$ and $\pi_{\mathrm{grid}}^{i,\mathrm{NN}}(\boldsymbol{x})$ is the estimated conditional density of $T = i/B$ given $\boldsymbol{X} = \boldsymbol{x}$. This structure is analogous to a classification network by removing the last layer which outputs the class with highest softmax score. Estimation of conditional density at other $t$'s given $\boldsymbol{X} = \boldsymbol{x}$ are obtained via linear interpolation:

$$\pi^{\mathrm{NN}}(t \mid \boldsymbol{x}) = \pi_{\mathrm{grid}}^{t_1,\mathrm{NN}}(\boldsymbol{x}) + B\left(\pi_{\mathrm{grid}}^{t_2,\mathrm{NN}}(\boldsymbol{x}) - \pi_{\mathrm{grid}}^{t_1,\mathrm{NN}}(\boldsymbol{x})\right)(t - t_1), \quad \text{where} \quad t_1 = \lfloor Bt \rfloor, t_2 = \lceil Bt \rceil.$$

This estimator $\pi^{\mathrm{NN}}$ is continuous with respect to $t$ for any given $\boldsymbol{x}$ and we finally rescale it to yield a valid density, i.e., $\pi^{\mathrm{NN}}(t \mid \boldsymbol{x}) \geq 0$, $\forall t, \boldsymbol{x}$ and $\int_{t=0}^{1} \pi^{\mathrm{NN}}(t \mid \boldsymbol{x}) dt = 1$, $\forall \boldsymbol{x}$.

There are other options to estimate the conditional density. Popular methods include the mixture density network (Bishop (1994)), the kernel mixture network (Ambrogioni et al. (2017)) and normalizing flows (Rezende & Mohamed (2015), Dinh et al. (2016), Trippe & Turner (2018)). Here treatment levels are bounded, and thus Gaussian mixtures are not applicable. In current estimator, the linear interpolation for estimating conditional density on non-grid points can be replaced by kernel smoothing, which is more computationally intensive due to the calculation of normalizing constant. Other techniques including smoothness regularization and data normalization (Rothfuss et al. (2019)) can be implemented to further enhance the performance. However, density estimation is not the main focus of this paper. Thus, for simplicity, in all experiments we use the aforementioned method without other techniques and it works quite well on datasets we tried.

### 3.3 TRAINING

Notice that our model requires $\pi^{\text{NN}}$ to extract good latent features $z$ as the input for $\mu^{\text{NN}}$ to predict. This can be achieved by training $\pi^{\text{NN}}$ to estimate the conditional density, which motivates us to train $\pi^{\text{NN}}$ and $\mu^{\text{NN}}$ simultaneously by minimizing the following loss:

$$\mathcal{L}[\mu^{\text{NN}}, \pi^{\text{NN}}] = \frac{1}{n} \sum_{i=1}^{n} \left(y_i - \mu^{\text{NN}}(t_i, \boldsymbol{x}_i)\right)^2 - \frac{\alpha}{n} \sum_{i=1}^{n} \log(\pi^{\text{NN}}(t_i \mid \boldsymbol{x}_i)). \tag{1}$$

In loss (1), the first term measures the prediction loss from $\mu^{\text{NN}}$. The second term measures the loss from $\pi^{\text{NN}}$ and is the negative log likelihood. And $\alpha$ controls the relative weights of the two losses.

Denote $\hat{\mu}, \hat{\pi}$ as the optimal solution of the above empirical risk minimization problem (1). After getting $\hat{\mu}$, one can estimate $\psi(\cdot)$ by $\hat{\psi}(\cdot) = \frac{1}{n} \sum_{i=1}^{n} \hat{\mu}(\cdot, \boldsymbol{x}_i)$. The correctness of this naive estimator relies on whether the truth $\mu$ is in the function space defined by the neural network model. However, we can plug $\hat{\mu}$ and $\hat{\pi}$ into the non-parametric estimating equation (to be introduced later) to obtain a doubly robust estimator of $\psi(t)$. In this way, we are able to produce an (asymptotically) correct estimator if any one of $\mu$ or $\pi$ is in the model space of neural network. The next section discusses challenges in obtaining such a doubly robust estimator and provides our solution.

## 4 FUNCTIONAL TARGETED REGULARIZATION

In this section we improve upon the previous method by utilizing semiparametric theory on doubly robust estimators. Doubly robust estimators are built upon $\hat{\pi}(t \mid \boldsymbol{x})$ and $\hat{\mu}(t, \boldsymbol{x})$, and it yields a consistent estimator for $\psi$ even if one of them is inconsistent. When both $\hat{\pi}(t \mid \boldsymbol{x})$ and $\hat{\mu}(t, \boldsymbol{x})$ are consistent, a doubly robust estimator leads to faster rates of convergence. Here our task is to estimate the whole ADRF curve, which comes with additional challenges. First, we need to find a doubly robust estimator of $\psi(t_0)$ for any $t_0 \in [0, 1]$.

### 4.1 DOUBLY ROBUST ESTIMATOR

Before we proceed, we define the following quantity:

$$\zeta_{t_0}(Y, \boldsymbol{X}, T, \pi, \mu, \psi) = q_{t_0}(Y, \boldsymbol{X}, T, \mu, \pi) + \mu(t_0, \boldsymbol{X}) - \psi(t_0),$$

$$\text{where} \quad q_{t_0}(Y, \boldsymbol{X}, T, \mu, \pi) = \delta(T - t_0) \frac{Y - \mu(T, \boldsymbol{X})}{\pi(T \mid \boldsymbol{X})}.$$

The following theorem serves as the basis for our subsequent estimators:

**Theorem 1.** *Under assumption 1 and assume that $\hat{\pi}(t \mid \boldsymbol{x}) \geq c > 0$ for all $\boldsymbol{x} \in \mathcal{X}$ and $t \in \mathcal{T}$. For any $t_0 \in \mathcal{T}$, $\zeta_{t_0}$ is the efficient influence function for $\psi(t_0)$. Moreover, $\zeta_{t_0}$ is doubly robust in the sense that*

$$\mathbb{P}\zeta_{t_0}(Y, \boldsymbol{X}, T, \hat{\pi}, \hat{\mu}, \psi) = 0$$

*if either $\hat{\pi} = \pi$ or $\hat{\mu} = \mu$. Further, if $\|\hat{\pi} - \pi\|_\infty = O_p(r_1(n))$ and $\|\hat{\mu} - \mu\|_\infty = O_p(r_2(n))$, we have*

$$\sup_{t_0 \in \mathcal{T}} |\mathbb{P}\zeta_{t_0}(Y, \boldsymbol{X}, T, \hat{\pi}, \hat{\mu}, \psi)| = O_p(r_1(n)r_2(n)).$$

Theorem 1 shows that for any $t_0 \in \mathcal{T}$, $\mathbb{P}\left(q_{t_0}(Y, \boldsymbol{X}, T, \hat{\mu}, \hat{\pi}) + \hat{\mu}(t_0, \boldsymbol{X})\right)$ is a doubly robust estimator for $\psi(t_0)$. Under some mild assumptions, one way to obtain a doubly robust estimator of $\psi$ is to utilize the two-stage procedure from Kennedy et al. (2017) by regressing

$$\frac{Y - \hat{\mu}(T, \boldsymbol{X})}{\hat{\pi}(T \mid \boldsymbol{X})} \int_{\mathcal{X}} \hat{\pi}(T \mid \boldsymbol{x}) d\mathbb{P}_n(\boldsymbol{x}) + \hat{\mu}(T, \boldsymbol{X}) \tag{2}$$

on $T$ using any nonparametric regression methods like kernel regression or spline regression.

## 4.2 TARGETED REGULARIZATION FOR INFERRING A FINITE DIMENSIONAL QUANTITY

However, as discussed in Shi et al. (2019), when estimating the term $\mathbb{P}\left(q_{t_0}(Y, \boldsymbol{X}, T, \hat{\mu}, \hat{\pi})\right)$, the $\hat{\pi}(T \mid \boldsymbol{X})$ in the denominator might make the finite sample estimator unstable, especially in cases where Assumption 1(a) is nearly violated. Targeted regularization is proposed by Shi et al. (2019) to solve this issue. The key intuition of targeted regularization is to learn $\hat{\mu}$ and $\hat{\pi}$ such that $\mathbb{P}\left(q_{t_0}(Y, \boldsymbol{X}, T, \hat{\mu}, \hat{\pi})\right) \approx 0$ and thus the estimation of this term is no more needed. In the binary treatment case where $\mathcal{T} = \{0, 1\}$, if we want to estimate $\psi(1)$, which is a single quantity, targeted regularization simultaneously optimizes over $\mu^{\mathrm{NN}}$, $\pi^{\mathrm{NN}}$ and an extra scalar perturbation parameter $\epsilon$ using the following loss

$$\mathcal{L}_{\mathrm{TR}}[\mu^{\mathrm{NN}}, \pi^{\mathrm{NN}}, \epsilon] = \mathcal{L}[\mu^{\mathrm{NN}}, \pi^{\mathrm{NN}}] + \beta \mathcal{R}_{\mathrm{TR}}[\mu^{\mathrm{NN}}, \pi^{\mathrm{NN}}, \epsilon], \tag{3}$$

$$\text{where } \mathcal{R}_{\mathrm{TR}}[\mu^{\mathrm{NN}}, \pi^{\mathrm{NN}}, \epsilon] = \frac{1}{n} \sum_{i=1}^{n} \left(y_i - \mu^{\mathrm{NN}}(t_i, \boldsymbol{x}_i) - \epsilon \frac{t_i}{\pi^{\mathrm{NN}}(1 \mid \boldsymbol{x}_i)}\right)^2,$$

with $\mathcal{L}[\mu^{\mathrm{NN}}, \pi^{\mathrm{NN}}]$ defined in (1). Assume the complexity of the function space of $\mu^{\mathrm{NN}}$ and $\pi^{\mathrm{NN}}$ is finite and since the complexity of the function space of the introduced perturbation $\epsilon$ is also finite, we have

$$\mathbb{P}\left[q_1\left(Y, \boldsymbol{X}, T, \hat{\mu}_{\mathrm{TR}}, \hat{\pi}\right)\right] = \mathbb{P}\left[q_1\left(Y, \boldsymbol{X}, T, \hat{\mu}_{\mathrm{TR}}, \hat{\pi}\right)\right] + \frac{1}{2}\frac{\partial}{\partial \epsilon} \mathcal{R}_{\mathrm{TR}}[\hat{\mu}, \hat{\pi}, \epsilon] \mid_{\epsilon=\hat{\epsilon}}$$
$$= (\mathbb{P} - \mathbb{P}_n)\left(q_1(Y, \boldsymbol{X}, T, \hat{\mu}, \hat{\pi}) - \hat{\epsilon}\delta(T - t_0)/\hat{\pi}^2(T \mid \boldsymbol{X})\right) = o_p(1),$$

where $\hat{\mu}_{\mathrm{TR}}(t, \boldsymbol{x}) := \hat{\mu}(t, \boldsymbol{x}) + \hat{\epsilon}\frac{t}{\hat{\pi}(t \mid \boldsymbol{x})}$ and $(\hat{\mu}, \hat{\pi}, \hat{\epsilon})$ is the minimizer of (3). Notice that the first equality holds because at the convergence of the optimization, $\frac{\partial}{\partial \epsilon} \mathcal{R}_{\mathrm{TR}}[\hat{\mu}, \hat{\pi}, \epsilon] \mid_{\epsilon=\hat{\epsilon}} = 0$. And the last equality is by uniform concentration inequality. This implies that $\frac{1}{n} \sum_{i=1}^{n} \hat{\mu}_{\mathrm{TR}}(1, \boldsymbol{x})$ is a doubly robust estimator for $\psi(1)$ and for this estimator, no conditional density estimator presents at denominator and thus it has more stable finite sample performance.

## 4.3 FUNCTIONAL TARGETED REGULARIZATION FOR INFERRING THE WHOLE ADRF

Notice that in loss (3), the scalar $\epsilon$ is associated with a scalar quantity for inference. One can generalize targeted regularization to estimate a $d$ dimensional vector by using $d$ separate $\epsilon$'s (see Theorem 3 in the Appendix). Generalizing to a curve, however, is more challenging. We need to optimize over a function $\epsilon : \mathcal{T} \to \mathbb{R}$ where $\epsilon(\cdot)$ is the perturbation associated with $\psi(\cdot)$. Optimizing over the function space of all mappings from $\mathcal{T}$ to $\mathbb{R}$ is not feasible in practice, and its high complexity will lead to overfitting.

Our solution is to utilize the smoothness of $\mu$ and $\pi$ (Prichard & Gillam, 1971; Schneider et al., 1993; Threlfall & English, 1999), which allows us to use splines $\{\varphi_k\}_{k=1}^{K_n}$ with $K_n$ basis functions to approximate $\epsilon(\cdot)$. Here the subscript $n$ in $K_n$ denotes that the number of basis functions might change with the sample size $n$. Define $\epsilon_n(\cdot) = \sum_{k=1}^{K_n} \alpha_k \varphi_k(\cdot)$. We use the following loss with Functional Targeted Regularization (FTR).

$$\mathcal{L}_{\mathrm{FTR}}[\mu^{\mathrm{NN}}, \pi^{\mathrm{NN}}, \epsilon_n] = \mathcal{L}[\mu^{\mathrm{NN}}, \pi^{\mathrm{NN}}] + \beta_n \mathcal{R}_{\mathrm{FTR}}[\mu^{\mathrm{NN}}, \pi^{\mathrm{NN}}, \epsilon_n] \tag{4}$$

$$\text{where } \mathcal{R}_{\mathrm{FTR}}[\mu^{\mathrm{NN}}, \pi^{\mathrm{NN}}, \epsilon_n] = \frac{1}{n} \sum_{i=1}^{n} \left(y_i - \mu^{\mathrm{NN}}(t_i, \boldsymbol{x}_i) - \frac{\epsilon_n(t_i)}{\pi^{\mathrm{NN}}(t_i \mid \boldsymbol{x}_i)}\right)^2.$$

Here $\mathcal{R}_{\mathrm{FTR}}$ denotes the FTR term and $\beta_n \to 0$ when $n \to \infty$.

**Remark 2 (On $\beta_n$).** *The targeted regularization proposed by Shi et al. (2019) uses a fixed $\beta$. However, using fixed $\beta$ might lead to the estimator constructed by targeted regularization no more consistent when $\mu^{NN}$ is mis-specified, which means the estimator is no more doubly robust. To overcome this issue, we make a slight change on $\beta$ by allowing $\beta$ to depend on $n$. Specifically, we find that once $\beta_n = o(1)$, we are able to make sure that targeted regularization gives doubly robust estimator. See discussion at Remark 4 and Appendix A.1 for more details.*

Demonstrating the asymptotic correctness of FTR is more challenging than analyzing traditional targeted regularization. One reason is that we no more have $\frac{\partial}{\partial \epsilon} \mathcal{R}_{\text{TR}}[\hat{\mu}, \hat{\pi}, \epsilon] \mid_{\epsilon = \hat{\epsilon}} = 0$. With some additional efforts, we will establish convergence rate for our estimator using loss (4) in Theorem 2. Before we proceed, let us pause a bit and introduce some definitions, which will be used in the main theorem. Denote $\hat{\mu}$, $\hat{\pi}$ and $\hat{\epsilon}_n$ as the minimizer of (4). We use $\bar{\pi}$ and $\bar{\mu}$ to denote fixed functions to which $\hat{\pi}$ and $\hat{\mu}$ converge in the sense that $\|\hat{\pi} - \bar{\pi}\|_{\infty} = o_p(1)$ and $\|\hat{\mu} - \bar{\mu}\|_{\infty} = o_p(1)$. We define $g_t : \mathcal{X} \to \mathbb{R}, \boldsymbol{x} \mapsto \mu^{NN}(\boldsymbol{x}, t)$. We denote $\mathcal{G}, \mathcal{Q}, \mathcal{U}$ as the function space in which $g_t$, $\mu^{NN}$, $\pi^{NN}$ lies. We denote $\mathcal{B}_{K_n}$ as the closed linear span of basis $\boldsymbol{\varphi}^{K_n} = \{\varphi_k\}_{k=1}^{K_n}$.

The key intuition of the asymptotic correctness of FTR is that: once $\pi^{NN}$ and $\pi$ are uniformly upper/lower bounded and some other weak regularization conditions hold, we can show that $\|\hat{\epsilon}_n(\cdot) - \epsilon^*(\cdot)\|_{L^2} = o_p(1)$ where $\epsilon^*(\cdot) := \mathbb{E}\left[(Y - \bar{\mu})/\bar{\pi} \mid T = \cdot\right]/\mathbb{E}\left[\bar{\pi}^{-2} \mid T = \cdot\right]$. And thus letting $\hat{\mu}_{\text{FTR}} := \hat{\mu} + \hat{\epsilon}_n/\hat{\pi}$, we have

$$\mathbb{P}\left[q_{t_0}\left(Y, \boldsymbol{X}, T, \hat{\mu}_{\text{FTR}}, \hat{\pi}\right)\right] = \mathbb{P}\left((Y - \hat{\mu}_{\text{FTR}})/\hat{\pi} \mid T = t_0\right)\pi(t_0)$$
$$\approx \left[\mathbb{E}\left((Y - \bar{\mu} - \epsilon^*/\bar{\pi})/\bar{\pi} \mid T = t_0\right)\right]\pi(t_0) = 0.$$

**Assumption 2.** *We consider the following assumptions:*

*(i) There exists constant $c > 0$ such that for any $t \in \mathcal{T}$, $\boldsymbol{x} \in \mathcal{X}$, and $\pi^{NN} \in \mathcal{U}$, we have $1/c \leq \pi^{NN}(t \mid \boldsymbol{x}) \leq c$, $1/c \leq \pi(t \mid \boldsymbol{x}) \leq c$, $\|\mathcal{Q}\|_{\infty} \leq c$ and $\|\mu\|_{\infty} \leq c$.*

*(ii) $Y = \mu(\boldsymbol{X}, T) + V$ where $\mathbb{E}V = 0$, $V \perp \boldsymbol{X}$, $V \perp T$, and $V$ follows sub-Gaussian distribution.*

*(iii) $\pi$, $\mu$, $\pi^{NN}$ and $\mu^{NN}$ have bounded second derivatives for any $\pi^{NN} \in \mathcal{Q}$ and $\mu^{NN} \in \mathcal{U}$.*

*(iv) Either $\bar{\pi} = \pi$ or $\bar{\mu} = \mu$. And $Rad_n(\mathcal{G})$, $Rad_n(\mathcal{Q})$, $Rad_n(\mathcal{U}) = O\left(n^{-1/2}\right)$.*

*(v) $\mathcal{B}_{K_n}$ equals the closed linear span of B-spline with equally spaced knots, fixed degree, and dimension $K_n \asymp n^{1/6}$.*

**Theorem 2.** *Under Assumption 1 and 2, let $\widehat{\psi}(\cdot) := \frac{1}{n}\sum_{i=1}^{n}\left(\hat{\mu}(\boldsymbol{x}_i, \cdot) + \frac{\hat{\epsilon}_n(\cdot)}{\hat{\pi}(\cdot \mid \boldsymbol{x}_i)}\right)$, we have*

$$\|\widehat{\psi} - \psi\|_{L^2} = O_p\left(n^{-1/3}\sqrt{\log n} + r_1(n)r_2(n)\right).$$

*where $\|\hat{\pi} - \pi\|_{\infty} = O_p(r_1(n))$ and $\|\hat{\mu} - \mu\|_{\infty} = O_p(r_2(n))$.*

**Remark 3.** *In Theorem 2, assumption (i), (iii) and the first half of (v) are weak and standard conditions for establishing convergence rate of spline estimators (Huang et al., 2003; 2004). Assumption (ii) bounds the tail behavior of $V$. The second half of (v) restricts the growth rate of $K_n$, which is a typical assumption (Huang et al., 2003; 2004) but with different rate in order to obtain uniform bound. The first half of assumption (iv) states that at least one of $\hat{\mu}, \hat{\pi}$ should be consistent. The second half of assumption (iv) considers the complexity of model space, and is a common assumption for problems with nuisance functions (Kennedy et al., 2017).*

**Remark 4.** *We want to point out that adding targeted regularization does not affect the limit of $\hat{\mu}$ and $\hat{\pi}$ in large sample asymptotics. That is, the limit of $\hat{\mu}$ and $\hat{\pi}$ using loss (4) will be the same as using loss (1). We refer the reader to Appendix A.1 for a more detailed discussion and proof.*

Notice that our proof for Theorem 2 can also be adapted for analyzing modified one-step TMLE (Van Der Laan & Rubin, 2006). With very similar assumptions, we could obtain double robustness and the same consistency rate for TMLE estimator.

Theorem 2 guarantees that if we appropriately control the model complexity, under some mild assumptions, the estimator $\widehat{\psi}$ from targeted regularization is doubly robust, and when both $\hat{\pi}$ and $\hat{\mu}$ are consistent, the rate of convergence of $\widehat{\psi}$ to the truth is faster than the individual convergence rate of $\hat{\pi}$ or $\hat{\mu}$. Thus, using targeted regularization theoretically helps us obtain a better estimator of $\psi$.

| Dataset | Model | Naive | Doubly Robust | TMLE | TR |
|---------|-------|-------|---------------|------|-----|
| Simulation | Dragonnet | $0.045 \pm 0.00094$ | $0.026 \pm 0.0012$ | $0.037 \pm 0.00086$ | $0.028 \pm 0.00088$ |
| | Drnet | $0.042 \pm 0.00090$ | $0.023 \pm 0.0011$ | $0.035 \pm 0.00083$ | $0.027 \pm 0.00086$ |
| | Vcnet | $0.018 \pm 0.00098$ | $0.022 \pm 0.0013$ | $0.016 \pm 0.00082$ | $0.014 \pm 0.00091$ |
| IHDP | Dragonnet | $0.350 \pm 0.016$ | $0.307 \pm 0.016$ | $0.252 \pm 0.0087$ | $0.208 \pm 0.0072$ |
| | DRnet | $0.316 \pm 0.016$ | $0.274 \pm 0.014$ | $0.274 \pm 0.018$ | $0.230 \pm 0.0086$ |
| | Vcnet | $0.189 \pm 0.013$ | $0.190 \pm 0.013$ | $0.148 \pm 0.010$ | $0.117 \pm 0.0085$ |
| News | Dragonnet | $0.180 \pm 0.0081$ | $0.155 \pm 0.0057$ | $0.179 \pm 0.0080$ | $0.149 \pm 0.0051$ |
| | DRnet | $0.183 \pm 0.0084$ | $0.141 \pm 0.0054$ | $0.183 \pm 0.0083$ | $0.114 \pm 0.0041$ |
| | Vcnet | $0.028 \pm 0.0011$ | $0.023 \pm 0.0013$ | $0.028 \pm 0.0010$ | $0.024 \pm 0.0009$ |

Table 1: Experiment result comparing neural network based methods. TR refers to targeted regularization. Numbers reported are AMSE of testing data based on 100 repeats for Simulation and IHDP and 20 repeats for News, and numbers after $\pm$ are the estimated standard deviation of the average value.

## 5 RELATED WORK

**The Varying Coefficient Structure.** Varying coefficient (linear) model is first proposed as an extension of linear model (Hastie & Tibshirani, 1993; Fan et al., 1999) and is usually used for modeling longitudinal data (Huang et al., 2004; Zhang & Wang, 2015; Li et al., 2017; Ye et al., 2019). The key motivation of varying coefficient model is a dimension reduction technique that avoids the curse of dimensionality for statistical estimation. Different from existing models, our varying coefficient structure is applied on a complex neural network model with a different motivation of enhancing the expressiveness of the treatment effect. Besides, building a hierarchical structure on the network parameter is also explored by the HyperNetwork (Stanley et al., 2009; Ha et al., 2016). Hypernetworks provide an abstraction that mimics the biology structure: the relationship between a genotype (the hypernetwork), and a phenotype (the main network). The weight of the main network is also a function of a latent embedding variable, which is learned with end-to-end training. Hypernetwork trains a much smaller network to generate the weights of a larger main network in order to reduce search space, while our network directly trains the main network whose weights are linear combinations of spline functions of treatment. Moreover our network is proposed in order to appropriately incorporate treatment into modelling, which is not touched upon in HyperNetwork.

**Neural Network Structure for Treatment Effect Estimation.** We refer readers to the introduction for the connections and comparisons with previous developments using feed forward neural network for treatment effect estimation. In addition to feed forward neural network, previous results also utilize other networks to learn treatment effects. For example, Yoon et al. (2018); Bica et al. (2020) estimated the causal effect via learning to generate the counterfactual. Louizos et al. (2017) learned the causal effect by learning deep variable models using variational autoencoder. Compared with our method, their approaches are mainly heuristic and do not provide theoretical guarantees for the asymptotic correctness of the estimator.

**Doubly Robustness, TMLE and Targeted Regularization.** Chernozhukov et al. (2017; 2018) developed theory for 'double machine learning' showing the convergence rate for doubly robust estimator. Despite its good asymptotic property, doubly robust estimators can be unstable due to the presence of conditional density estimator at denominator. Targeted Maximum Likelihood Estimation (TMLE) (Van der Laan & Rose, 2011) and targeted regularization (Shi et al., 2019) are then proposed to overcome this issue by introducing an extra perturbation parameter into the model. To the best of our knowledge, previous works on TMLE and targeted regularization focused on estimating a single quantity, such as $\psi(1) - \psi(0)$ in binary treatment (Shi et al., 2019; Van der Laan & Rose, 2011) or averaged treatment effect $\mathbb{E}\psi(t)$ for continuous treatment (Kennedy et al., 2017), while we give the first generalization of targeted regularization and TMLE for inferring the whole ADRF curve.

## 6 EXPERIMENTS

**Dataset.** Since the true ADRF are rarely available for real-world data, previous methods on treatment effect estimation often use synthetic/semi-synthetic data for empirical evaluation. Following this convention, we consider one synthetic and two semi-synthetic datasets: IHDP (Hill, 2011) and News (Newman, 2008). The synthetic dataset contains 500 training points and 200 testing points, with

| Method | Simulation | IHDP | News |
|---|---|---|---|
| Causal Forest | $0.043 \pm 0.0021$ | $0.97 \pm 0.034$ | $0.211 \pm 0.003$ |
| BART | $0.040 \pm 0.0013$ | $0.33 \pm 0.005$ | $0.066 \pm 0.003$ |
| GPS | $0.028 \pm 0.0016$ | $0.67 \pm 0.025$ | $0.022 \pm 0.001$ |
| VCNet+TR | $0.014 \pm 0.0009$ | $0.12 \pm 0.009$ | $0.024 \pm 0.001$ |

Table 2: Comparison of VCNet against non-neural-network based baselines. Reported AMSE are averaged over 100 experiments for simulation and IHDP, and 20 experiments for News. Numbers after $\pm$ are estimated standard deviation of the average AMSE.

the detailed generating scheme included in the Appendix. IHDP contains binary treatment with 747 observations on 25 covariates, and News consists of 3000 randomly sampled news items from the NY Times corpus (Newman, 2008). Both IHDP and News are widely used benchmarking datasets for binary treatment effect estimation, but here we focus on continuous treatment and thus we need to generate the continuous treatment as well as outcome by ourselves. The generating scheme is in the Appendix. For IHDP and news, we randomly split into training set (67%) and testing set (33%).

**Baselines and Settings.** For neural network baselines, we compare against Dragonnet (Shi et al., 2019) and DRNet (Schwab et al., 2019). We improve upon the original Dragonnet and DRNet by (a) using separate heads for $T$ in different blocks for Dragonnet, and (b) adding a conditional density estimation head for DRNet, since it has been suggested by Shi et al. (2019) that adding a conditional density estimation head improves the performance. For non-neural-network baselines, we consider causal forest (Wager & Athey, 2018), Bayesian Additive Regression Tree (BART) (Chipman et al., 2010), and GPS (Imbens, 2000).

For VCNet, we use truncated polynomial basis with degree 2 and two knots at $\{1/3, 2/3\}$ (thus altogether 5 basis). Dragonnet and DRNet use 5 blocks and thus the model complexity of neural-network models are the same. In practice we may vary the degree and number of knots in VCNet, here the choice is made simply for fair comparison against Dragonnet and DRNet, ensuring the number of parameters of the compared models is the same. The other hyper-parameters of each method on each data are tuned on 20 separate tuning sets. Due to space limit, we refer readers to the Appendix A.4 for more details on experimental settings.

**Estimator and Metrics.** To evaluate the effectiveness of targeted regularization, for all neural-network methods we implement four versions: naive version (with conditional density estimator head, trained using loss (1)), doubly robust version (Kennedy et al., 2017) by regressing (2) on treatment with $\hat{\mu}, \hat{\pi}$ trained using loss (1), TMLE (Van Der Laan & Rubin, 2006) version with initial estimator trained using loss (1), and TR version trained using loss (4). For non-neural-network based models, we use the usual estimator. For evaluation metric, following Schwab et al. (2019), we use the average mean squared error (AMSE) on test set, where AMSE $= \frac{1}{S} \sum_{s=1}^{S} \int_{\mathcal{T}} [\hat{\psi}_s(t) - \psi(t)]^2 \pi(t) dt$ and $\hat{\psi}_s(t)$ is the estimated $\psi(t)$ in the $s$-th simulation.

**Results.** Table 1 compares neural network based methods. Comparing results in each column, we observe a performance boost from the varying coefficient structure. Comparing results in each row, we find that naive versions consistently perform the worst, targeted regularization often achieves the best performance, whereas performance of doubly robust estimator and TMLE varies across datasets. In Table 2, we compare our approach with traditional statistical models. We observe that in simulation and IHDP, VCNet + targeted regularization outperforms baselines by a large margin. In News, its performance is close to the best one. The implementation can be found in an open source repository[1].

## 7 CONCLUSION

This work proposes a novel varying coefficient network and generalizes targeted regularization to a continuous curve. We provide theorems showing its consistency and double robustness. Experiments show that VCNet structure and targeted regularization boost performance independently and when used together, it improves over existing methods by a large margin.

---

[1]https://github.com/lushleaf/varying-coefficient-net-with-functional-tr

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

# A APPENDIX

## A.1 ON THE CONSISTENCY OF $\bar{\mu}$ AND $\bar{\pi}$

We show that adding targeted regularization does not affect the limit of $\hat{\mu}$ and $\hat{\pi}$ in large sample asymptotics. That is, the limit of $\hat{\mu}$ and $\hat{\pi}$ using loss (4) will be the same as using loss (1).

Denote

$$\mathbb{P}\ell(\mu^{\mathrm{NN}}, \pi^{\mathrm{NN}}) = \mathbb{P}\left[(y - \mu^{\mathrm{NN}})^2 + \alpha \log \pi^{\mathrm{NN}}(t \mid \boldsymbol{x})\right],$$

$$\mathbb{P}_n\ell(\mu^{\mathrm{NN}}, \pi^{\mathrm{NN}}) = \frac{1}{n}\sum_{i=1}^n \left[(y_i - \mu^{\mathrm{NN}}(t_i, \boldsymbol{x}_i))^2 + \alpha \pi^{\mathrm{NN}}(t_i \mid \boldsymbol{x}_i)\right].$$

**Lemma 1.** *Suppose that $(\mu', \pi')$ is the minimizer of loss $\mathbb{P}\ell(\mu^{NN}, \pi^{NN})$ and $(\hat{\mu}, \hat{\pi}, \hat{\epsilon}_n)$ is the minimizer of $\mathcal{L}_{FTR}$, then we have*

$$\mathbb{P}\ell(\hat{\mu}, \hat{\pi}) - \mathbb{P}\ell(\mu', \pi') = o(1) + O_p(n^{-1/2}).$$

*Proof.* We have

$$
\begin{aligned}
\mathbb{P}\ell(\hat{\mu}, \hat{\pi}) - \mathbb{P}\ell(\mu', \pi') &\leq \mathbb{P}_n\ell(\hat{\mu}, \hat{\pi}) - \mathbb{P}_n\ell(\mu', \pi') + |(\mathbb{P} - \mathbb{P}_n)\,\ell(\hat{\mu}, \hat{\pi})| + |(\mathbb{P} - \mathbb{P}_n)\,\ell(\mu', \pi')| \\
&\stackrel{(a)}{\leq} \mathbb{P}_n\ell(\hat{\mu}, \hat{\pi}) - \mathbb{P}\ell(\mu', \pi') + O_p(n^{-1/2}) \\
&= (\mathbb{P}_n\ell(\hat{\mu}, \hat{\pi}) + \beta_n \mathcal{R}_{\mathrm{FTR}}[\hat{\mu}, \hat{\pi}, \hat{\epsilon}_n]) - (\mathbb{P}\ell(\mu', \pi') + \beta_n \mathcal{R}_{\mathrm{FTR}}[\mu', \pi', 0]) + O_p(n^{-1/2}) \\
&\quad + \beta_n\left(\mathcal{R}_{\mathrm{FTR}}[\mu', \pi', 0] - \mathcal{R}_{\mathrm{FTR}}[\hat{\mu}, \hat{\pi}, \hat{\epsilon}_n]\right) \\
&\stackrel{(b)}{\leq} \beta_n\left(\mathcal{R}_{\mathrm{FTR}}[\mu', \pi', 0] - \mathcal{R}_{\mathrm{FTR}}[\hat{\mu}, \hat{\pi}, \hat{\epsilon}_n]\right) + O_p(n^{-1/2}) \\
&\leq \beta_n \mathcal{R}_{\mathrm{FTR}}[\mu', \pi', 0] + O_p(n^{-1/2}) \\
&\stackrel{(c)}{=} o(1) + O_p(n^{-1/2}),
\end{aligned}
$$

where $(a)$ follows from uniform concentration inequality using the fact that $\mu^{\mathrm{NN}}, \pi^{\mathrm{NN}}$ is uniformly bounded, $\mathrm{Rad}_n(\mathcal{Q}), \mathrm{Rad}_n(\mathcal{U}) = O(n^{-1/2})$ and the Lipschitz constant of $\log(x)$ is bounded when $x \in [1/c, c]$ for some finite $c$. $(b)$ follows from the fact that $(\hat{\mu}, \hat{\pi}, \hat{\epsilon}_n)$ is the minimizer of the empirical risk (with FTR). $(c)$ follows from the fact that

$$
\begin{aligned}
\mathcal{R}[\mu', \pi', 0] &= \frac{1}{n}\sum_{i=1}^n (y_i - \mu')^2 \\
&= \mathbb{E}(y - \mu')^2 + O_p(n^{-1/2}) \\
&= \mathbb{E}(V^2) + \mathbb{E}(\mu - \mu')^2 + O_p(n^{-1/2}) \\
&= O(1).
\end{aligned}
$$

$\square$

Now we prove that

$$\|\hat{\mu} - \mu'\|_{L^2} + \|\hat{\pi} - \pi'\|_{L^2} = o_p(1). \tag{5}$$

For simplicity, we ignore the unidentifiability of neural network parameterization and assume $(\mu', \pi')$ is the unique minimizer in the sense that for any $\epsilon > 0$, there exists $\eta(\epsilon) > 0$ such that

$$\inf_{\|\mu^{\mathrm{NN}} - \mu'\|_{L^2} + \|\pi^{\mathrm{NN}} - \pi'\|_{L^2} > \epsilon} \left[\mathbb{P}\ell(\mu^{\mathrm{NN}}, \pi^{\mathrm{NN}}) - \mathbb{P}\ell(\mu', \pi')\right] > \eta(\epsilon). \tag{6}$$

If Equation (5) is not true, then there exists $s > 0$ such that for any $N > 0$, there exists $n > N$ such that $\|\hat{\mu} - \mu'\|_{L^2} + \|\hat{\pi} - \pi'\|_{L^2} \geq s$. From Lemma 1, we know that $\mathbb{P}\ell(\hat{\mu}, \hat{\pi}) - \mathbb{P}\ell(\mu', \pi') \leq \eta(s)$ when $n$ is sufficiently large. Thus, there exists $n_0$ such that

$$\|\hat{\mu} - \mu'\|_{L^2} + \|\hat{\pi} - \pi'\|_{L^2} \geq s, \quad \mathbb{P}\ell(\hat{\mu}, \hat{\pi}) - \mathbb{P}\ell(\mu', \pi') \leq \eta(s),$$

which contradicts (6).

## A.2 TECHNICAL PROOFS

In this section, we prove the two main theorems (Theorem 1 and Theorem 2) and give some additional results which are mentioned briefly in the main text.

### A.2.1 NOTATIONS AND DEFINITIONS

We denote $\delta_{t_0}$ as the Dirac measure centered on $t_0$ and recall that $\delta(\cdot)$ denotes the Dirac delta function. We use $a_n \lesssim b_n$ to denote that $a_n \leq C b_n$ for some $C > 0$ for all sufficiently large $n$. We denote $\mathbf{1}_n = (1, 1, \cdots, 1)^T \in \mathbb{R}^n$. For any function $f$ and function spaces $\mathcal{F}_1, \mathcal{F}_2$, we write $\mathcal{F}_1 + \mathcal{F}_2 = \{f_1 + f_2 : f_1 \in \mathcal{F}_1, f_2 \in \mathcal{F}_2\}$, $\mathcal{F}_1 \mathcal{F}_2 = \{f_1 f_2 : f_1 \in \mathcal{F}_1, f_2 \in \mathcal{F}_2\}$, $f\mathcal{F} = \{fh : h \in \mathcal{F}\}$, $f \circ \mathcal{F} = \{f \circ h : h \in \mathcal{F}\}$, and $\mathcal{F}^a = \{f^a : f \in \mathcal{F}\}$, $\forall a \in \mathbb{R}$.

We define

$$\check{\epsilon}_n(\cdot) = \mathbb{P}\left[(Y - \hat{\mu}_n)/\hat{\pi}_n \mid T = \cdot\right]/\mathbb{P}\left[\hat{\pi}_n^{-2} \mid T = \cdot\right],$$

where $(\hat{\mu}_n, \hat{\pi}_n, \hat{\epsilon}_n)$ is the minimizer of loss (4). We denote $\hat{\epsilon}_n(\cdot) = \sum_{k=1}^K \hat{\alpha}_k \varphi_k(\cdot)$ as the spline regression estimator of $\epsilon(\cdot)$. With some slight abuse of notation, we denote

$$\boldsymbol{\varphi}^{K_n}(t) = (\varphi_1(t), \varphi_2(t), \cdots, \varphi_{K_n}(t))^T \in \mathbb{R}^{K_n},$$

$$B_n = \left(\boldsymbol{\varphi}^{K_n}(t_1), \cdots, \boldsymbol{\varphi}^{K_n}(t_n)\right)^T \in \mathbb{R}^{n \times K_n}.$$

We define

$$\Pi_n = \operatorname{diag}\left(\hat{\pi}_n(t_1 \mid \boldsymbol{x}_1), \hat{\pi}_n(t_2 \mid \boldsymbol{x}_2), \cdots, \hat{\pi}_n(t_n \mid \boldsymbol{x}_n)\right),$$

$$\tilde{\Pi}_n = \operatorname{diag}\left(\left[\mathbb{P}\left(\hat{\pi}_n^{-2}(T \mid \boldsymbol{X}) \mid T = t_1\right)\right]^{-1/2}, \cdots, \left[\mathbb{P}\left(\hat{\pi}_n^{-2}(T \mid \boldsymbol{X}) \mid T = t_n\right)\right]^{-1/2}\right).$$

We define

$$\boldsymbol{Z}_n = (z_1, z_2, \cdots, z_n)^T \in \mathbb{R}^n \quad \text{where} \quad z_i = \frac{y_i - \hat{\mu}_n(t_i, \boldsymbol{x}_i)}{\hat{\pi}_n(t_i \mid \boldsymbol{x}_i)},$$

$$\tilde{\boldsymbol{Z}}_n = (\tilde{z}_1, \tilde{z}_2, \cdots, \tilde{z}_n)^T \in \mathbb{R}^n \quad \text{where} \quad \tilde{z}_i = \mathbb{P}\left(\frac{Y - \hat{\mu}_n(T, \boldsymbol{X})}{\hat{\pi}_n(T \mid \boldsymbol{X})} \mid T = t_i\right).$$

Notice that

$$\hat{\boldsymbol{\alpha}} = \left(B_n^T \Pi_n^{-2} B_n\right)^{-1} B_n^T \boldsymbol{Z}_n,$$

and we denote

$$\tilde{\boldsymbol{\alpha}} = \left(B_n^T \Pi_n^{-2} B_n\right)^{-1} B_n^T \Pi_n^{-2} \tilde{\Pi}_n^2 \tilde{\boldsymbol{Z}}_n.$$

### A.2.2 USEFUL LEMMAS

This section gives lemmas which are used in our proofs of main theorems.

Recall that Theorem 1 consists of two parts: the efficient influence function of $\psi(t_0)$ and its double robustness. Notice that $\psi(t_0)$ can be written as a special case of a more general parameter $\Gamma = \int_{\mathcal{T}} \gamma(t; \mathbb{P}_T) \psi(t) d\mathbb{P}_T(t)$ (see Remark 5). Here $\gamma(t; \mathbb{P}_T)$ is a function of $t$ which depends on the probability measure $\mathbb{P}_T$. For brevity we also write $\gamma(t) := \gamma(t; \mathbb{P}_T)$ when the corresponding $\mathbb{P}_T$ is the true probability measure of treatment $T$. The following Lemma gives the efficient influence function of $\Gamma$.

**Lemma 2.** *The efficient influence function for $\Gamma$ is*

$$\zeta(Y, \boldsymbol{X}, T, \pi, \mu, \Gamma) = \gamma(T)\xi(Y, \boldsymbol{X}, T, \pi, \mu) - \Gamma + \int_{\mathcal{T}} \mu(t, \boldsymbol{X})\gamma(t)d\mathbb{P}_T(t) + \frac{\gamma'_\varepsilon(T)}{l'_\varepsilon(T; 0)}\int_{\mathcal{X}} \mu(T, \boldsymbol{x})d\mathbb{P}(\boldsymbol{x})$$

$$- \int_{\mathcal{T}}\int_{\mathcal{X}} \gamma(t)\mu(t, x)d\mathbb{P}(x)d\mathbb{P}_T(t) - \mathbb{E}_T\left[\frac{\gamma'_\varepsilon(T)}{l'_\varepsilon(T; 0)}\int_{\mathcal{X}} \mu(T, \boldsymbol{x})d\mathbb{P}(\boldsymbol{x})\right],$$

*where* $\xi(Y, \boldsymbol{X}, T, \pi, \mu, \Gamma) = \frac{Y - \mu(T, \boldsymbol{X})}{\pi(T \mid \boldsymbol{X})}\int_{\mathcal{X}} \pi(T \mid \boldsymbol{x})d\mathbb{P}(\boldsymbol{x}) + \int_{\mathcal{X}} \mu(T, \boldsymbol{x})d\mathbb{P}(\boldsymbol{x})$, $\gamma'_\varepsilon(t) = \frac{d\gamma(t; \mathbb{P}_{T,\varepsilon})}{d\varepsilon}\big|_{\varepsilon=0}$, $l'_\varepsilon(t; 0) = \frac{\partial \log \mathbb{P}_{T,\varepsilon}(t)}{\partial \varepsilon}\big|_{\varepsilon=0}$ *and* $\mathbb{P}_{T,\varepsilon}$ *is a parametric submodel with parameter* $\varepsilon \in \mathbb{R}$ *and* $\mathbb{P}_{T,0}(\cdot) = \mathbb{P}_T(\cdot)$.

**Remark 5.** *Setting $\gamma(\cdot) = \frac{d\delta_{t_0}(\cdot)}{d\mathbb{P}_T(\cdot)}$, we get $\Gamma = \psi(t_0)$. Setting $\gamma(\cdot) = 1$, we get $\Gamma = \int_{\mathcal{T}} \psi(t)d\mathbb{P}_T(t)$, which is the average outcome under a randomized trial and is a quantity of interest in its own right (Kennedy et al., 2017). Setting $\gamma(\cdot) = \frac{d\delta_1(\cdot)}{d\mathbb{P}_T(\cdot)} - \frac{d\delta_0(\cdot)}{d\mathbb{P}_T(\cdot)}$, we get $\Gamma = \psi(1) - \psi(0)$, which is the average treatment effect under binary treatment setting.*

Recall that in Theorem 2, $\hat{\epsilon}_n$ is the spline regression estimator of $\epsilon$. In order to establish the convergence rate of our final estimator $\hat{\psi}$, we need to establish the convergence rate of $\hat{\epsilon}_n$ first.

**Lemma 3.** *Under assumptions in Theorem 2, we have*

$$\|\hat{\epsilon}_n - \check{\epsilon}_n\|_{L^2} = O_p\left(n^{-1/3}\sqrt{\log n}\right).$$

**Remark 6.** *For fixed objective function, the convergence rate of the B-spline estimator to the truth is a standard result (Huang et al., 2004; 2003), which is $O_p(n^{-2/5})$ when choosing $K_n \asymp n^{-1/5}$. However, Lemma 3 gives a uniform bound on a class of functions in order to deal with the fact that $\hat{\pi}_n$ and $\hat{\mu}_n$ are NOT fixed and dependent on the observations. The bound is thus of a larger order $n^{-1/3}\sqrt{\log n}$ when choosing the optimal $K_n \asymp n^{-1/6}$.*

**Lemma 4.** *Under assumption in Theorem 2, there exist positive constants $M_1$ and $M_2$ such that except on an event whose probability tends to zero, all the eigenvalues of $(K_n/n) B_n^T \Pi_n^{-2} B_n$ fall between $M_1$ and $M_2$, and consequently, $(K_n/n) B_n^T \Pi_n^{-2} B_n$ is invertible.*

**Lemma 5.** *Assume $\|\mathcal{F}_1\|_\infty < \infty$ and $\|\mathcal{F}_2\|_\infty < \infty$, we have*

$$Rad_n(\mathcal{F}_1\mathcal{F}_2) \le \frac{1}{2}(Rad_n(\mathcal{F}_1) + Rad_n(\mathcal{F}_2))(\|\mathcal{F}_1\|_\infty + \|\mathcal{F}_2\|_\infty).$$

### A.2.3 PROOF OF THEOREM 1

*Proof.* Denote $\gamma(t; \mathbb{P}_T) = \frac{d\delta_{t_0}(t)}{d\mathbb{P}_T(t)}$. Then the efficient influence function $\zeta_{t_0}$ of $\psi(t_0)$ is obtained by plugging the definition of $\gamma(t; \mathbb{P}_T)$ in Lemma 2 and some simplifications using

$$\frac{\gamma'_\varepsilon(t; \mathbb{P}_T)}{l'_\varepsilon(t; 0)} = -\gamma(t; \mathbb{P}_T). \tag{7}$$

So here we only need to prove that the efficient influence function is doubly robust. We have

$$
\begin{aligned}
&\mathbb{P}\zeta_{t_0}(Y, \boldsymbol{X}, T, \hat{\pi}, \hat{\mu}, \Gamma) \\
&= \mathbb{P}\left(\delta(T - t_0)\frac{Y - \hat{\mu}(T, \boldsymbol{X})}{\hat{\pi}(T \mid \boldsymbol{X})} + \hat{\mu}(t_0, \boldsymbol{X}) - \psi(t_0)\right) \\
&\overset{(a)}{=} \int \pi(\boldsymbol{x})\pi(t \mid \boldsymbol{x})\delta(t - t_0)\frac{\mu(t, \boldsymbol{x}) - \hat{\mu}(t, \boldsymbol{x})}{\hat{\pi}(t \mid \boldsymbol{x})}dxdt + \int \hat{\mu}(t_0, \boldsymbol{x})\pi(\boldsymbol{x})d\boldsymbol{x} - \int \mu(t_0, \boldsymbol{x})\pi(\boldsymbol{x})d\boldsymbol{x} \\
&\overset{(b)}{=} \int_{\mathcal{X}} \left(\frac{\pi(t_0 \mid \boldsymbol{x})}{\hat{\pi}(t_0 \mid \boldsymbol{x})} - 1\right)(\mu(t_0, \boldsymbol{x}) - \hat{\mu}(t_0, \boldsymbol{x}))\,d\mathbb{P}(\boldsymbol{x}),
\end{aligned}
\tag{8}
$$

where (a) follows from iterated expectation, (b) follows from $\pi(\boldsymbol{x} \mid t) = \pi(t \mid \boldsymbol{x})\pi(\boldsymbol{x})/\pi(t)$. From the last line of Equation (8), it is obvious that the desired conclusions hold. $\qquad\square$

### A.2.4 PROOF OF THEOREM 2

*Proof.* First, from condition (i), we have

$$
\left\| \hat{\epsilon}(\cdot) \int_{\mathcal{X}} \frac{1}{\hat{\pi}(\cdot \mid \mathbf{X})} d\mathbb{P}_n(\mathbf{X}) - \mathbb{P}\left[ \delta(T - \cdot) \frac{Y - \hat{\mu}(T, \mathbf{X})}{\hat{\pi}(T \mid \mathbf{X})} \right] \right\|_{L^2}
$$

$$
= \left\| \hat{\epsilon}(\cdot) \int_{\mathcal{X}} \frac{1}{\hat{\pi}(\cdot \mid \mathbf{X})} d\mathbb{P}_n(\mathbf{X}) - \pi(\cdot)\mathbb{P}\left( \frac{Y - \hat{\mu}_n(\mathbf{X}, T)}{\hat{\pi}_n(\mathbf{X}, T)} \mid T = \cdot \right) \right\|_{L^2}
$$

$$
\leq \left\| (\hat{\epsilon}(\cdot) - \check{\epsilon}(\cdot)) \int_{\mathcal{X}} \frac{1}{\hat{\pi}(\cdot \mid \mathbf{X})} d\mathbb{P}_n(\mathbf{X}) \right\|_{L^2} + \left\| \check{\epsilon}(\cdot) \left( \int_{\mathcal{X}} \frac{1}{\hat{\pi}(\cdot \mid \mathbf{X})} d\mathbb{P}_n(\mathbf{X}) - \pi(\cdot)\mathbb{P}\left( \frac{1}{\hat{\pi}^2(\cdot \mid \mathbf{X})} \mid T = \cdot \right) \right) \right\|_{L^2}
$$

$$
\lesssim \| \hat{\epsilon} - \check{\epsilon} \|_{L^2} + \left\| \check{\epsilon}(\cdot) \left( \int_{\mathcal{X}} \frac{1}{\hat{\pi}(\cdot \mid \mathbf{X})} d\left( \mathbb{P}_n - \mathbb{P} \right)(\mathbf{X}) \right) + \check{\epsilon}(\cdot) \left( \int_{\mathcal{X}} \frac{1}{\hat{\pi}(\cdot \mid \mathbf{X})} d\mathbb{P}(\mathbf{X}) - \pi(\cdot)\mathbb{P}\left( \frac{1}{\hat{\pi}^2(\cdot \mid \mathbf{X})} \mid T = \cdot \right) \right) \right\|_{L^2}
$$

$$
\lesssim \| \hat{\epsilon} - \check{\epsilon} \|_{L^2} + \left\| \check{\epsilon}(\cdot) \left( \int_{\mathcal{X}} \frac{1}{\hat{\pi}(\cdot \mid \mathbf{X})} d\left( \mathbb{P}_n - \mathbb{P} \right)(\mathbf{X}) \right) \right\|_{L^2}
$$

$$
+ \left\| \mathbb{P}\left( \frac{\mu(\mathbf{X}, T) - \hat{\mu}_n(\mathbf{X}, T)}{\hat{\pi}_n(\mathbf{X}, T)} \mid T = \cdot \right) \int_{\mathcal{X}} \frac{\hat{\pi}(\cdot \mid \mathbf{X}) - \pi(\cdot \mid \mathbf{X})}{\hat{\pi}(\cdot \mid \mathbf{X})} \frac{1}{\hat{\pi}(\cdot \mid \mathbf{X})} d\mathbb{P}(\mathbf{X}) \right\|_{L^2}
$$

$$
\stackrel{(a)}{=} O_p\left( n^{-1/3}\sqrt{\log n} + r_1(n)r_2(n) \right).
$$

$$\tag{9}$$

where $(a)$ follows from Lemma 3, which says $\| \hat{\epsilon}_n - \check{\epsilon}_n \|_{L^2} = O_p(n^{-1/3}\sqrt{\log n})$.

From generalization bound and condition (iv), we know

$$
\sup_{t_0 \in [0,1]} \left| \frac{1}{n} \sum_{i=1}^{n} \hat{\mu}_n(x_i, t_0) - \mathbb{P}\hat{\mu}_n(X, t_0) \right| = \sup_{t_0 \in [0,1]} |\mathbb{P}_n \hat{g}_{t_0}(X) - \mathbb{P}\hat{g}_{t_0}(X)| = O_p(n^{-1/2}).
$$

Thus,

$$
\left\| \frac{1}{n} \sum_{i=1}^{n} \hat{\mu}_n(x_i, \cdot) - \mathbb{P}\hat{\mu}_n(X, \cdot) \right\|_{L^2} = O_p(n^{-1/2})
\tag{10}
$$

Recall that Theorem 1 says that if $\sup_{t \in [0,1]} \sup_{\mathbf{X} \in \mathcal{X}} |\hat{\pi}_n(t \mid \mathbf{X}) - \pi(t \mid \mathbf{X})| = O_p(r_1(n))$, $\sup_{t \in [0,1]} \sup_{\mathbf{X} \in \mathcal{X}} |\hat{\mu}_n(t, \mathbf{X}) - Q(t, \mathbf{X})| = O_p(r_2(n))$, then

$$
\sup_{t_0 \in [0,1]} \left| \mathbb{P}\left[ \delta(T - t_0) \frac{Y - \hat{\mu}_n(T, \mathbf{X})}{\hat{\pi}_n(T \mid \mathbf{X})} + \hat{\mu}_n(t_0, \mathbf{X}) \right] - \psi(t_0) \right| = O_p(r_1(n)r_2(n)).
\tag{11}
$$

Combining Equation (9), Equation (10) and Equation (11), using triangle inequality, we have

$$
\left\| \hat{\epsilon}(\cdot) \int_{\mathcal{X}} \frac{1}{\hat{\pi}(\cdot \mid \mathbf{x})} d\mathbb{P}_n(\mathbf{x}) + \frac{1}{n} \sum_{i=1}^{n} \hat{\mu}_n(x_i, \cdot) - \psi(\cdot) \right\|_{L^2} = O_p(n^{-1/3}\sqrt{\log n} + r_1(n)r_2(n)).
$$

So if we set

$$
\hat{\psi}(t_0) = \hat{\epsilon}(t_0) \int_{\mathcal{X}} \frac{1}{\hat{\pi}(t_0 \mid \mathbf{x})} d\mathbb{P}_n(\mathbf{x}) + \frac{1}{n} \sum_{i=1}^{n} \hat{\mu}_n(x_i, t_0) = \frac{1}{n} \sum_{i=1}^{n} \left( \hat{\mu}(\mathbf{x}_i, t_0) + \frac{\hat{\epsilon}(t_0)}{\hat{\pi}(t_0 \mid \mathbf{x}_i)} \right)
$$

we have

$$
\|\hat{\psi} - \psi\|_{L^2} = O_p\left( n^{-1/3}\sqrt{\log n} + r_1(n)r_2(n) \right).
$$

$\square$

### A.2.5 PROOF OF LEMMA 2

*Proof.* The proof follows Kennedy et al. (2017). Denote

$$
\Gamma(\varepsilon) = \int_{\mathcal{T}} \gamma_\varepsilon(t) \int_{\mathcal{X}} \int_{\mathcal{Y}} y\pi(y \mid \mathbf{x}, t; \varepsilon)\pi(\mathbf{x}; \varepsilon)\pi(t; \varepsilon) dy d\mathbf{x} dt,
$$

where we write $\gamma_\varepsilon(\cdot) := \gamma(\cdot\,; \mathbb{P}_{T,\varepsilon})$ for brevity. Then, by definition, the efficient influence function for $\Gamma$ is the unique function $\zeta(Y, \boldsymbol{X}, T)$ such that

$$\Gamma'_\varepsilon(0) = \mathbb{E}\left(\zeta(Y, \boldsymbol{X}, T)\ell'_\varepsilon(Y, \boldsymbol{X}, T; 0)\right) \tag{12}$$

where $\ell'_\varepsilon(y, \boldsymbol{x}, t; 0) = \frac{d \log \mathbb{P}_{Y,\boldsymbol{X},T,\varepsilon}(y,\boldsymbol{x},t)}{d\varepsilon}|_{\varepsilon=0}$, $\mathbb{P}_{Y,\boldsymbol{X},T,\varepsilon}(y, \boldsymbol{x}, t)$ is a parametric submodel with parameter $\varepsilon \in \mathbb{R}$, and $\mathbb{P}_{Y,\boldsymbol{X},T,0}(y, \boldsymbol{x}, t) = \mathbb{P}_{Y,\boldsymbol{X},T}(y, \boldsymbol{x}, t)$, and $\Gamma'_\epsilon(0) = d\Gamma(\varepsilon)/d\varepsilon|_{\varepsilon=0}$. We have

$$\Gamma'_\varepsilon(0) = \int_{\mathcal{T}} \gamma(t) \int_{\mathcal{X}} \int_{\mathcal{Y}} y\left[\pi'_\varepsilon(y \mid \boldsymbol{x}, t; 0)\pi(\boldsymbol{x}) + \pi(y \mid \boldsymbol{x}, t)\pi'_\varepsilon(\boldsymbol{x}; 0)\right] \pi(t) dy d\boldsymbol{x} dt$$

$$+ \int_{\mathcal{T}} \gamma_\varepsilon(t) \int_{\mathcal{X}} \int_{\mathcal{Y}} y\pi(y \mid \boldsymbol{x}, t; \epsilon)\pi(\boldsymbol{x}; \varepsilon)\pi'_\varepsilon(t; 0) dy d\boldsymbol{x} dt$$

$$+ \int_{\mathcal{T}} \gamma'_\varepsilon(t) \int_{\mathcal{X}} \int_{\mathcal{Y}} y\pi(y \mid \boldsymbol{x}, t; \varepsilon)\pi(\boldsymbol{x}; \varepsilon)\pi(t; \varepsilon) dy d\boldsymbol{x} dt$$

$$= I_1 + I_2 + I_3.$$

From $\ell'_\varepsilon(y \mid \boldsymbol{x}, t; 0) = \pi'_\varepsilon(y \mid \boldsymbol{x}, t; 0)/\pi(y \mid \boldsymbol{x}, t; 0)$ and definition of $\psi(t)$, we have

$$I_1 := \int_{\mathcal{T}} \gamma(t) \int_{\mathcal{X}} \int_{\mathcal{Y}} y\left[\ell'_\varepsilon(y \mid \boldsymbol{x}, t; 0)\pi(y \mid \boldsymbol{x}, t)\pi(\boldsymbol{x}) + \pi(y \mid \boldsymbol{x}, t)\ell'_\varepsilon(\boldsymbol{x}; 0)\pi(\boldsymbol{x})\right] \pi(t) dy d\boldsymbol{x} dt$$

$$= \int_{\mathcal{T}} \gamma(t) \left[\mathbb{E}_{\boldsymbol{X}}\mathbb{E}_{Y|\boldsymbol{X},T}\left(y\ell'_\varepsilon(y \mid \boldsymbol{x}, t; 0)\right) + \mathbb{E}_{\boldsymbol{X}}\left(\mu(\boldsymbol{x}, t)\ell'_\epsilon(\boldsymbol{x}; 0)\right)\right] \pi(t) dt,$$

$$I_2 := \int_{\mathcal{T}} \gamma(t)\psi(t)\pi'_\varepsilon(t; 0) dt = \int_{\mathcal{T}} \gamma(t)\psi(t)\ell'_\varepsilon(t; 0)\pi(t) dt,$$

$$I_3 := \int_{\mathcal{T}} \gamma'_\varepsilon(t)\psi(t)\pi(t; 0) dt.$$

Thus, we have

$$\Gamma'_\varepsilon(0) = \int_{\mathcal{T}} \left[\gamma(t)\mathbb{E}_{\boldsymbol{X}}\mathbb{E}_{Y|\boldsymbol{X},T}\left(y\ell'_\epsilon(y \mid \boldsymbol{x}, t; 0)\right) + \mathbb{E}_{\boldsymbol{X}}\left(\mu(\boldsymbol{x}, t)\ell'_\epsilon(\boldsymbol{x}; 0)\right) + \gamma(t)\psi(t)\ell'_\epsilon(t; 0) + \gamma'_\epsilon(t)\psi(t)\right] \pi(t) dt. \tag{13}$$

Meanwhile, for the right hand size term $\mathbb{E}_{\boldsymbol{X},T,Y}\left(\zeta(Y, \boldsymbol{X}, T)\ell'_\varepsilon(Y, \boldsymbol{X}, T; 0)\right)$ in Equation (12), from

$$\ell'_\varepsilon(Y, \boldsymbol{X}, T; 0) = \ell'_\varepsilon(Y|\boldsymbol{X}, T; 0) + \ell'_\varepsilon(\boldsymbol{X}, T; 0),$$

where $\ell'_\varepsilon(Y|\boldsymbol{X}, T; 0)$ and $\ell'_\varepsilon(\boldsymbol{X}, T; 0)$ are defined similar to $\ell'_\varepsilon(Y, \boldsymbol{X}, T; 0)$, we know that

$$\mathbb{E}_{\boldsymbol{X},T,Y}\left(\zeta(Y, \boldsymbol{X}, T)\ell'_\varepsilon(Y, \boldsymbol{X}, T; 0)\right) = \mathbb{E}_{\boldsymbol{X},T,Y}\left(\zeta(Y, \boldsymbol{X}, T)\ell'_\varepsilon(Y|\boldsymbol{X}, T; 0)\right) + \mathbb{E}_{\boldsymbol{X},T,Y}\left(\zeta(Y, \boldsymbol{X}, T)\ell'_\varepsilon(\boldsymbol{X}, T; 0)\right). \tag{14}$$

Now we bound each term in Equation (14) separately. Recall that

$$\zeta(Y, \boldsymbol{X}, T, \pi, \mu) = \gamma(T)\frac{Y - \mu(T, \boldsymbol{X})}{\pi(T \mid \boldsymbol{X})} \int_{\mathcal{X}} \pi(T \mid \boldsymbol{x}) d\mathbb{P}(\boldsymbol{x}) + \gamma(T) \int_{\mathcal{X}} \mu(T, \boldsymbol{x}) d\mathbb{P}(\boldsymbol{x}) - \Gamma$$

$$+ \int_{\mathcal{T}} \mu(t, \boldsymbol{X})\gamma(t) d\mathbb{P}_T(t) + \frac{\gamma'_\varepsilon(T)}{l'_\varepsilon(T; 0)} \int_{\mathcal{X}} \mu(T, \boldsymbol{x}) d\mathbb{P}(\boldsymbol{x})$$

$$- \int_{\mathcal{T}} \int_{\mathcal{X}} \gamma(t)\mu(t, x) dp(x) d\mathbb{P}_T(t) - \mathbb{E}_T\left[\frac{\gamma'_\varepsilon(T)}{l'_\varepsilon(T; 0)} \int_{\mathcal{X}} \mu(T, \boldsymbol{x}) d\mathbb{P}(\boldsymbol{x})\right].$$

Thus, for the first term in Equation (14), we have

$$\mathbb{E}_{\boldsymbol{X},T,Y}\left(\zeta(Y, \boldsymbol{X}, T)\ell'_\varepsilon(Y|\boldsymbol{X}, T; 0)\right)$$

$$= \mathbb{E}_{\boldsymbol{X},T}\left[\mathbb{E}_{Y|\boldsymbol{X},T}\left(\zeta(Y, \boldsymbol{X}, T)\ell'_\varepsilon(Y|\boldsymbol{X}, T; 0)\right)\right]$$

$$\overset{(a)}{=} \mathbb{E}_{\boldsymbol{X},T}\mathbb{E}_{Y|\boldsymbol{X},T}\left[\left(\gamma(T)\frac{Y}{\pi(T \mid \boldsymbol{X})} \int_{\mathcal{X}} \pi(T \mid \boldsymbol{x}) d\mathbb{P}(\boldsymbol{x})\right) \ell'_\varepsilon(Y|\boldsymbol{X}, T; 0)\right]$$

$$\overset{(b)}{=} \int_{T \times \mathcal{X}}\left[\gamma(t)\frac{\mathbb{E}_{Y|\boldsymbol{x},t}\left[Y\ell'_\varepsilon(Y|\boldsymbol{x}, t; 0)\right]}{\pi(t \mid \boldsymbol{x})}\left(\int_{\mathcal{X}} \pi(t \mid \boldsymbol{x}) d\mathbb{P}(\boldsymbol{x})\right) p(\boldsymbol{x})\pi(t \mid \boldsymbol{x})\right] dt d\boldsymbol{x} \tag{15}$$

$$= \int_{\mathcal{T} \times \mathcal{X}} \gamma(t)\mathbb{E}_{Y|\boldsymbol{x},t}\left[Y\ell'_\varepsilon(Y|\boldsymbol{x}, t; 0)\right] p(t)p(\boldsymbol{x}) dt d\boldsymbol{x}$$

$$= \int_{\mathcal{T}} \gamma(t)\mathbb{E}_{\boldsymbol{x}}\left[\mathbb{E}_{Y|\boldsymbol{x},t}\left[Y\ell'_\varepsilon(Y|\boldsymbol{x}, t; 0)\right]\right] p(t) dt,$$

where $(a)$ follows from the fact that $\mathbb{E}_{Y|\boldsymbol{X},T}\left(\ell'_\epsilon(Y|\boldsymbol{X},T;0)\right)=0$, $(b)$ uses law of iterated expectations.

For the second term in Equation (14), we have

$$
\begin{aligned}
&\mathbb{E}_{\boldsymbol{X},T,Y}\left(\zeta(Y,\boldsymbol{X},T)\ell'_\varepsilon(\boldsymbol{X},T;0)\right)\\
&\overset{(a)}{=}\mathbb{E}_{\boldsymbol{X},T,Y}\left[\left(\gamma(T)\frac{Y-\mu(T,\boldsymbol{X})}{\pi(T\mid\boldsymbol{X})}\int_{\mathcal{X}}\pi(T\mid\boldsymbol{x})d\mathbb{P}(\boldsymbol{x})\right)\ell'_\varepsilon(\boldsymbol{X},T;0)\right]\\
&\quad+\mathbb{E}_{\boldsymbol{X},T}\left(\gamma(T)\int_{\mathcal{X}}\mu(T,\boldsymbol{x})d\mathbb{P}(\boldsymbol{x})+\frac{\gamma'_\varepsilon(T)}{l'_\varepsilon(T;0)}\int_{\mathcal{X}}\mu(T,\boldsymbol{x})d\mathbb{P}(\boldsymbol{x})\right)\left(\ell'_\varepsilon(\boldsymbol{X}|T;0)+\ell'_\varepsilon(T;0)\right)\\
&\quad+\mathbb{E}_{\boldsymbol{X},T}\left[\left(\int_{\mathcal{T}}\mu(t,\boldsymbol{X})\gamma(t)d\mathbb{P}(t)\right)\left(\ell'_\varepsilon(T|\boldsymbol{X};0)+\ell'_\varepsilon(\boldsymbol{X};0)\right)\right]\\
&\quad+\mathbb{E}_{\boldsymbol{X},T}\left[\left(-\int_{\mathcal{T}}\gamma(t)\psi(t)d\mathbb{P}(t)-\mathbb{E}_T\left[\frac{\gamma'_\varepsilon(T)}{l'_\varepsilon(T;0)}\int_{\mathcal{X}}\mu(T,\boldsymbol{x})d\mathbb{P}(\boldsymbol{x})\right]-\Gamma\right)\left(\ell'_\varepsilon(T,\boldsymbol{X};0)\right)\right]\\
&\overset{(b)}{=}\mathbb{E}_T\left[\gamma(T)\int_{\mathcal{X}}\mu(T,\boldsymbol{x})d\mathbb{P}(\boldsymbol{x})\ell'_\varepsilon(T;0)\right]+\mathbb{E}_T\left[\frac{\gamma'_\varepsilon(T)}{l'_\varepsilon(T;0)}\int_{\mathcal{X}}\mu(T,\boldsymbol{x})d\mathbb{P}(\boldsymbol{x})\ell'_\varepsilon(T;0)\right]\\
&\quad+\mathbb{E}_{\boldsymbol{X}}\left[\int_{\mathcal{T}}\mu(t,\boldsymbol{X})\gamma(t)d\mathbb{P}(t)\ell'_\varepsilon(\boldsymbol{X};0)\right]\\
&\overset{(c)}{=}\int_{\mathcal{T}}\gamma(t)\psi(t)\ell'_\varepsilon(t;0)p(t)dt+\int_{\mathcal{T}}\mathbb{E}_{\boldsymbol{X}}\left[\mu(t,\boldsymbol{X})\ell'_\varepsilon(\boldsymbol{X};0)\right]\gamma(t)d\mathbb{P}(t)+\int_{\mathcal{T}}\gamma'_\varepsilon(t)\psi(t)p(t)dt
\end{aligned}
\tag{16}
$$

where $(a)$ follows from the fact that

$$
\ell'_\varepsilon(\boldsymbol{X},T;0)=\ell'_\varepsilon(\boldsymbol{X}|T;0)+\ell'_\varepsilon(T;0)=\ell'_\varepsilon(T|\boldsymbol{X};0)+\ell'_\varepsilon(\boldsymbol{X};0),
$$

(b) follows from the fact that

$$
\mathbb{E}_{Y|\boldsymbol{X},T}Y=\mu(T,\boldsymbol{X}),\quad\mathbb{E}_{T|\boldsymbol{X}}\ell'_\varepsilon(T|\boldsymbol{X};0)=\mathbb{E}_{\boldsymbol{X}|T}\ell'_\varepsilon(\boldsymbol{X}|T;0)=\mathbb{E}_{\boldsymbol{X},T}\ell'_\varepsilon(\boldsymbol{X},T;0)=0
$$

and law of iterated expectations, $(c)$ follows from the definition of $\psi(t)=\int_{\mathcal{X}}\mu(t,\boldsymbol{x})d\mathbb{P}(\boldsymbol{x})$.

Comparing Equation (15), Equation (16) against Equation (13), we immediately get

$$
\Gamma'_\varepsilon(0)=\mathbb{E}_{\boldsymbol{X},T,Y}\left(\zeta(Y,\boldsymbol{X},T)\ell'_\varepsilon(Y,\boldsymbol{X},T;0)\right),
$$

which implies that $\zeta(Y,\boldsymbol{X},T)$ is indeed an efficient influence function of $\Gamma$. $\qquad\square$

### A.2.6 PROOF OF LEMMA 3

*Proof.* This proof follows from Huang et al. (2004). Let us start with the following decomposition:

$$
\|\hat{\epsilon}_n-\check{\epsilon}_n\|_{L^2}\le\|\check{\epsilon}_n-\tilde{\epsilon}_n\|_{L^2}+\|\hat{\epsilon}_n-\tilde{\epsilon}_n\|_{L^2}
$$

where $\tilde{\epsilon}_n=\left(\boldsymbol{\varphi}^{K_n}(t)\right)^T\tilde{\boldsymbol{\alpha}}$. The first term $\|\check{\epsilon}_n-\tilde{\epsilon}_n\|_{L^2}$ is the bias and the second term $\|\hat{\epsilon}_n-\tilde{\epsilon}_n\|_{L^2}$ is the variance.

**Bound on bias term** Let $\check{\boldsymbol{\alpha}}\in\mathbb{R}^{K_n}$ be such that $\|\left(\check{\boldsymbol{\alpha}}\right)^T\boldsymbol{\varphi}^{K_n}-\check{\epsilon}_n\|_\infty=\inf_{f\in\mathcal{B}_{K_n}}\|f-\check{\epsilon}_n\|_\infty$. Then we have

$$
\begin{aligned}
\|\check{\epsilon}_n-\tilde{\epsilon}_n\|_{L^2}&=\|\check{\epsilon}_n-\left(\check{\boldsymbol{\alpha}}\right)^T\boldsymbol{\varphi}^{K_n}+\left(\check{\boldsymbol{\alpha}}\right)^T\boldsymbol{\varphi}^{K_n}-\tilde{\epsilon}_n\|_{L^2}\\
&\le\|\check{\epsilon}_n-\left(\check{\boldsymbol{\alpha}}\right)^T\boldsymbol{\varphi}^{K_n}\|_{L^2}+\|\left(\check{\boldsymbol{\alpha}}\right)^T\boldsymbol{\varphi}^{K_n}-\tilde{\epsilon}_n\|_{L^2}.
\end{aligned}
$$

By definition of $\check{\boldsymbol{\alpha}}$ and properties of B-spline space, we have a bound on the first term

$$
\|\check{\epsilon}_n-\left(\check{\boldsymbol{\alpha}}\right)^T\boldsymbol{\varphi}^{K_n}\|_{L^2}=O_p\left(\rho_n\right),
$$

where $\rho_n = \inf_{f \in \mathrm{Span}\{\boldsymbol{\varphi}^{K_n}\}} \sup_{t \in [0,1]} |\check{\epsilon}_n(t) - f(t)|$. Notice that the second term can also be bounded:

$$
\begin{aligned}
& \| (\check{\boldsymbol{\alpha}})^T \boldsymbol{\varphi}^{K_n} - \tilde{\epsilon}_n \|_{L^2} \\
& \overset{(a)}{\asymp} \| \check{\boldsymbol{\alpha}} - \tilde{\boldsymbol{\alpha}} \|_2 / \sqrt{K_n} \\
& = \| (B_n^T \Pi_n^{-2} B_n)^{-1} B_n^T \Pi_n^{-2} (B_n \check{\boldsymbol{\alpha}} - \tilde{\Pi}_n^2 \tilde{\boldsymbol{Z}}_n) \|_2 / \sqrt{K_n} \\
& \overset{(b)}{\asymp} \frac{K_n}{n} \| B_n^T \Pi_n^{-2} (B_n \check{\boldsymbol{\alpha}} - \tilde{\Pi}_n^2 \tilde{\boldsymbol{Z}}_n) \|_2 / \sqrt{K_n} \\
& \overset{(c)}{\asymp} \frac{\sqrt{K_n}}{n} \rho_n \sqrt{\mathbf{1}^T \Pi_n^{-2} B_n^T B_n \Pi_n^{-2} \mathbf{1}} \\
& \asymp \frac{\sqrt{K_n}}{n} \rho_n \sqrt{\sum_{k=1}^{K_n} \left( \sum_{i=1}^{n} \frac{\varphi_k(t_i)}{\hat{\pi}_n(t_i \mid \boldsymbol{x}_i)} \right)^2} \\
& \overset{(d)}{\asymp} \sqrt{K_n} \rho_n \sqrt{\sum_{k=1}^{K_n} \left( \frac{1}{n} \sum_{i=1}^{n} \varphi_k(t_i) \right)^2},
\end{aligned}
\tag{17}
$$

where (a) follows from properties of B-spline basis functions, (b) follows from Lemma 4, (c) follows from properties of B-spline space such that $\left\| B_n \check{\boldsymbol{\alpha}} - \tilde{\Pi}_n^2 \tilde{\boldsymbol{Z}}_n \right\|_\infty = O_p(\rho_n)$ because $(\check{\epsilon}_n(t_1), \check{\epsilon}_n(t_1), \cdots, \check{\epsilon}_n(t_1))^T = \tilde{\Pi}_n^2 \tilde{\boldsymbol{Z}}_n$, (d) is from the upper and lower boundedness of $\hat{\pi}_n$. Following proof of Lemma A.6 of Huang et al. (2004), for any $a > [\mathbb{E}\varphi_k(T)]^2 K_n$, we have

$$
\begin{aligned}
& \mathrm{Prob}\left( \sum_{k=1}^{K_n} \left( \frac{1}{n} \sum_{i=1}^{n} \varphi_k(t_i) \right)^2 > a \right) \\
& \overset{(a)}{\leq} \sum_{k=1}^{K_n} \mathrm{Prob}\left( \left| \frac{1}{n} \sum_{i=1}^{n} \varphi_k(t_i) \right| > \sqrt{\frac{a}{K_n}} \right) \\
& \leq \sum_{k=1}^{K_n} \mathrm{Prob}\left( \left| \frac{1}{n} \sum_{i=1}^{n} \varphi_k(t_i) - \mathbb{E}\varphi_k(T) \right| + |\mathbb{E}\varphi_k(T)| > \sqrt{\frac{a}{K_n}} \right) \\
& \leq \sum_{k=1}^{K_n} \mathrm{Prob}\left( \left| \frac{1}{n} \sum_{i=1}^{n} \varphi_k(t_i) - \mathbb{E}\varphi_k(T) \right| > \sqrt{\frac{a}{K_n}} - |\mathbb{E}\varphi_k(T)| \right) \\
& \overset{(b)}{\leq} 2 K_n \exp\left\{ -2n \left( \sqrt{a/K_n} - |\mathbb{E}\varphi_k(T)| \right)^2 \right\},
\end{aligned}
$$

where (a) uses union bound, (b) follows from Hoeffding's Inequality for bounded random variables. Since $\mathbb{E}\varphi_k(T) \asymp 1/K_n$, we can pick $a = 2[\mathbb{E}\varphi_k(T)]^2 K_n \asymp 1/K_n$ and thus, $\sum_{k=1}^{K_n} \left( \frac{1}{n} \sum_{i=1}^{n} \varphi_k(t_i) \right)^2 = O_p\left( \frac{1}{K_n} \right)$. Plugging into Equation (17), we get

$$
\| (\check{\boldsymbol{\alpha}})^T \boldsymbol{\varphi}^{K_n} - \tilde{\epsilon}_n \|_{L^2} = O_p(\rho_n),
$$

Thus, we can bound the bias term

$$
\| \check{\epsilon}_n - \tilde{\epsilon}_n \|_{L^2} = O_p(\rho_n).
$$

**Bound on variance term** From properties of B-spline space, we have

$$
\| \hat{\epsilon}_n - \tilde{\epsilon}_n \|_{L^2} \lesssim \| \hat{\boldsymbol{\alpha}} - \tilde{\boldsymbol{\alpha}} \|_2 / \sqrt{K_n}.
$$

Notice that

$$
\begin{aligned}
\| \hat{\boldsymbol{\alpha}} - \tilde{\boldsymbol{\alpha}} \|_2 & = \left\| (B_n^T \Pi_n^{-2} B_n)^{-1} B_n^T \left( \boldsymbol{Z}_n - \Pi_n^{-2} \tilde{\Pi}_n^2 \tilde{\boldsymbol{Z}}_n \right) \right\|_2 \\
& = \left\| (B_n^T \Pi_n^{-2} B_n)^{-1} B_n^T \left( \boldsymbol{Z}_n - \tilde{\boldsymbol{Z}}_n + \tilde{\boldsymbol{Z}}_n - \Pi_n^{-2} \tilde{\Pi}_n^2 \tilde{\boldsymbol{Z}}_n \right) \right\|_2 \\
& \leq \left\| (B_n^T \Pi_n^{-2} B_n)^{-1} B_n^T \left( \boldsymbol{Z}_n - \tilde{\boldsymbol{Z}}_n \right) \right\|_2 + \left\| (B_n^T \Pi_n^{-2} B_n)^{-1} B_n^T \left( \tilde{\boldsymbol{Z}}_n - \Pi_n^{-2} \tilde{\Pi}_n^2 \tilde{\boldsymbol{Z}}_n \right) \right\|_2
\end{aligned}
\tag{18}
$$

Control of the first term in Equation (18): denote $\boldsymbol{\delta} = (\delta_1, \cdots, \delta_n)^T := \boldsymbol{Z}_n - \tilde{\boldsymbol{Z}}_n$, then we have

$$
\left\| \left( B_n^T \Pi_n^{-2} B_n \right)^{-1} B_n^T \left( \boldsymbol{Z}_n - \tilde{\boldsymbol{Z}}_n \right) \right\|_2^2 = \left( \boldsymbol{Z}_n - \tilde{\boldsymbol{Z}}_n \right)^T B_n \left( B_n^T \Pi_n^{-2} B_n \right)^{-2} B_n^T \left( \boldsymbol{Z}_n - \tilde{\boldsymbol{Z}}_n \right)
$$

$$
\overset{(a)}{\asymp} \frac{K_n^2}{n^2} \boldsymbol{\delta}^T B_n B_n^T \boldsymbol{\delta} = \frac{K_n^2}{n^2} \left\| \sum_{i=1}^{n} \boldsymbol{\varphi}^{K_n}(t_i) \delta_i \right\|_2^2 = \frac{K_n^2}{n^2} \sum_{k=1}^{K_n} \left( \sum_{i=1}^{n} \varphi_k(t_i) \delta_i \right)^2
$$

$$
\leq K_n^2 \sum_{k=1}^{K_n} \sup_{\hat{\pi}, \hat{\mu}} \left( \frac{1}{n} \sum_{i=1}^{n} \varphi_k(t_i) \delta_i \right)^2
$$

$$(19)$$

where (a) is from Lemma 4. By definition we know

$$
\delta_i = \frac{y_i - \hat{\mu}_n(\boldsymbol{x}_i, t_i)}{\hat{\pi}(t_i \mid \boldsymbol{x}_i)} - \mathbb{P} \left( \frac{Y - \hat{\mu}_n(\boldsymbol{X}, T)}{\hat{\pi}(T \mid \boldsymbol{X})} \mid T = t_i \right)
$$

$$
= \frac{\mu(\boldsymbol{x}_i, t_i) - \hat{\mu}_n(\boldsymbol{x}_i, t_i)}{\hat{\pi}(t_i \mid \boldsymbol{x}_i)} - \mathbb{P} \left( \frac{\mu(\boldsymbol{X}, T) - \hat{\mu}_n(\boldsymbol{X}, T)}{\hat{\pi}(T \mid \boldsymbol{X})} \mid T = t_i \right) + \frac{v_i}{\hat{\pi}(t_i \mid \boldsymbol{x}_i)}
$$

$$
= u_i + \tilde{v}_i,
$$

where $u_i = \frac{\mu(\boldsymbol{x}_i, t_i) - \hat{\mu}_n(\boldsymbol{x}_i, t_i)}{\hat{\pi}(t_i \mid \boldsymbol{x}_i)} - \mathbb{P} \left( \frac{\mu(\boldsymbol{X}, T) - \hat{\mu}_n(\boldsymbol{X}, T)}{\hat{\pi}(T \mid \boldsymbol{X})} \mid T = t_i \right)$, $\mathbb{E}(u_i \mid t_i) = 0$ and $\tilde{v}_i = \frac{v_i}{\hat{\pi}(t_i \mid \boldsymbol{x}_i)}$, $\mathbb{E}(\tilde{v}_i \mid t_i, \boldsymbol{x}_i) = 0$. Thus, from union bound, we have

$$
\text{Prob} \left( \sum_{k=1}^{K_n} \sup_{\hat{\pi}, \hat{\mu}} \left( \frac{1}{n} \sum_{i=1}^{n} \varphi_k(t_i) \delta_i \right)^2 > a \right)
$$

$$
\leq \sum_{k=1}^{K_n} \text{Prob} \left( \sup_{\hat{\pi}, \hat{\mu}} \left( \frac{1}{n} \sum_{i=1}^{n} \varphi_k(t_i) \delta_i \right)^2 > \frac{a}{K_n} \right)
$$

$$
= \sum_{k=1}^{K_n} \text{Prob} \left( \sup_{\hat{\pi}, \hat{\mu}} \left| \frac{1}{n} \sum_{i=1}^{n} \varphi_k(t_i) \delta_i \right| > \sqrt{\frac{a}{K_n}} \right)
$$

$$
= \sum_{k=1}^{K_n} \text{Prob} \left( \sup_{\hat{\pi}, \hat{\mu}} \left| \frac{1}{n} \sum_{i=1}^{n} \varphi_k(t_i)(u_i + \tilde{v}_i) \right| > \sqrt{\frac{a}{K_n}} \right)
$$

$$
= \sum_{k=1}^{K_n} \text{Prob} \left( \sup_{\hat{\pi}, \hat{\mu}} \left| \frac{1}{n} \sum_{i=1}^{n} \varphi_k(t_i) u_i \right| > \frac{1}{2} \sqrt{\frac{a}{K_n}} \right) + \sum_{k=1}^{K_n} \text{Prob} \left( \sup_{\hat{\pi}, \hat{\mu}} \left| \frac{1}{n} \sum_{i=1}^{n} \varphi_k(t_i) \tilde{v}_i \right| > \frac{1}{2} \sqrt{\frac{a}{K_n}} \right).
$$

$$(20)$$

From Lemma 5, we know that

$$
\text{Rad}_n((\mathcal{Q} + \mu)\mathcal{U}^{-1}) \leq \frac{1}{2}(\|\mathcal{Q}\|_\infty + \|\mathcal{U}^{-1}\|_\infty) \left( \text{Rad}_n(\mathcal{Q}) + \text{Rad}_n(\mathcal{U}^{-1}) \right)
$$

$$
\overset{(a)}{\leq} \frac{1}{2}(\|\mathcal{Q}\|_\infty + \|\mathcal{U}^{-1}\|_\infty) \left( \text{Rad}_n(\mathcal{Q}) + \max \left( \frac{c^2}{2}, \frac{2}{(c - 1/c)^2} \right) \text{Rad}_n(\mathcal{U} - \frac{1}{2c}) + \frac{2c}{n} \right)
$$

$$
= O(n^{-1/2}),
$$

where (a) follows from plugging $h : x \mapsto \frac{1}{x - 1/2c} + 2c$ in Theorem 12(4) in Bartlett & Mendelson (2002). Similarly, write $\mathcal{A} = (\mathcal{Q} + \mu)\mathcal{U}^{-1}$, from Lemma 5, we have

$$
\text{Rad}_n(\varphi_k \mathcal{A}) \leq \frac{1}{2}(\|\varphi_k\|_\infty + \|\mathcal{A}\|_\infty)(\text{Rad}_n(\varphi_k) + \text{Rad}_n(\mathcal{A})) = O(n^{-1/2}).
$$

Thus, we bound the first term of (20) using

$$
\text{Prob} \left( \sup_{\hat{\pi}, \hat{\mu}} \left| \frac{1}{n} \sum_{i=1}^{n} \varphi_k(t_i) u_i \right| > \frac{1}{2} \sqrt{\frac{a}{K_n}} \right) \overset{(a)}{\lesssim} \frac{\mathbb{E} \left( \sup_{\hat{\pi}, \hat{\mu}} \left| \frac{1}{n} \sum_{i=1}^{n} \varphi_k(t_i) u_i \right| \right)}{\frac{1}{2} \sqrt{\frac{a}{K_n}}} \overset{(b)}{\asymp} \sqrt{\frac{K_n}{an}}, \quad (21)
$$

where (a) follows Markov Inequality, and (b) follows from the definition of Rademacher complexity.

We bound the second term of Equation (20) using union bound: for any $M_n > 0$,

$$\text{Prob}\left(\sup_{\hat{\pi},\hat{\mu}}\left|\frac{1}{n}\sum_{i=1}^{n}\varphi_k(t_i)\tilde{v}_i\right| > \frac{1}{2}\sqrt{\frac{a}{K_n}}\right)$$

$$\leq \text{Prob}\left(\sup_{\hat{\pi},\hat{\mu}}\left|\frac{1}{n}\sum_{i=1}^{n}\varphi_k(t_i)\tilde{v}_i\mathbb{I}(|v_i| > M_n)\right| > \frac{1}{4}\sqrt{\frac{a}{K_n}}\right) + \text{Prob}\left(\sup_{\hat{\pi},\hat{\mu}}\left|\frac{1}{n}\sum_{i=1}^{n}\varphi_k(t_i)\tilde{v}_i\mathbb{I}(|v_i| \leq M_n)\right| > \frac{1}{4}\sqrt{\frac{a}{K_n}}\right).$$

$$\tag{22}$$

We have from Markov Inequality that

$$\text{Prob}\left(\sup_{\hat{\pi},\hat{\mu}}\left|\frac{1}{n}\sum_{i=1}^{n}\varphi_k(t_i)\tilde{v}_i\mathbb{I}(|v_i| \leq M_n)\right| > \frac{1}{4}\sqrt{\frac{a}{K_n}}\right)$$

$$\lesssim \frac{\mathbb{E}\sup_{\hat{\pi},\hat{\mu}}\left|\frac{1}{n}\sum_{i=1}^{n}\varphi_k(t_i)/\hat{\pi}(t_i\mid \boldsymbol{x}_i)v_i\mathbb{I}(|v_i|\leq M_n)\right|}{\sqrt{a/K_n}} \overset{(a)}{\lesssim} \sqrt{\frac{K_n}{an}}M_n,$$

$$\tag{23}$$

where (a) follows from Lemma 5. Also we have

$$\text{Prob}\left(\sup_{\hat{\pi},\hat{\mu}}\left|\frac{1}{n}\sum_{i=1}^{n}\varphi_k(t_i)\tilde{v}_i\mathbb{I}(|v_i| > M_n)\right| > \frac{1}{4}\sqrt{\frac{a}{K_n}}\right)$$

$$\lesssim \frac{\mathbb{E}\sup_{\hat{\pi},\hat{\mu}}\left|\frac{1}{n}\sum_{i=1}^{n}\varphi_k(t_i)\tilde{v}_i\mathbb{I}(|v_i| > M_n)\right|}{\sqrt{a/K_n}}$$

$$\leq \frac{\mathbb{E}\sup_{\hat{\pi},\hat{\mu}}\frac{1}{n}\sum_{i=1}^{n}\frac{\varphi_k(t_i)}{\hat{\pi}(t_i|\boldsymbol{x}_i)}|v_i|\mathbb{I}(|v_i| > M_n)}{\sqrt{a/K_n}}$$

$$\lesssim \frac{\mathbb{E}\left[|v|\mathbb{I}(|v| > M_n)\right]}{\sqrt{a/K_n}}$$

$$\overset{(a)}{=} \frac{\int_0^\infty (1 - F_W(w))\,dw - \int_{-\infty}^0 F_W(w)dw}{\sqrt{a/K_n}}$$

$$\tag{24}$$

$$= \frac{\int_0^\infty \mathbb{P}(|v| \geq \max(M_n, w))dw}{\sqrt{a/K_n}}$$

$$\overset{(b)}{\lesssim} \frac{\int_0^\infty e^{-\sigma[\max(M_n,w)]^2}dw}{\sqrt{a/K_n}}$$

$$\leq \frac{\int_0^\infty e^{-\sigma[M_n+w]^2}dw}{\sqrt{a/K_n}}$$

$$\overset{(c)}{\lesssim} \frac{e^{-\sigma M^2}\sqrt{K_n}}{M_n}\frac{1}{\sqrt{a}},$$

where (a) uses the formula $\mathbb{E}W = \int_0^\infty (1-F(w))dw - \int_{-\infty}^0 F(w)dw$ and we set $W = |v|\mathbb{I}(|v| > M)$, (b) utilizes the fact that $v$ follows sub-Gaussian distribution, (c) uses Mills ratio.

Plugging Equation (21), (23), (24) into (20), and taking $M \asymp \sqrt{\log n}$, $a \asymp \frac{K_n \log n}{n}$, we get

$$\sum_{k=1}^{K_n}\sup_{\hat{\pi},\hat{\mu}}\left(\frac{1}{n}\sum_{i=1}^{n}\varphi_k(t_i)\delta_i\right)^2 = O_p\left(\frac{K_n \log n}{n}\right)$$

$$\tag{25}$$

which, when plugging back into Equation (19), gives

$$\left\|\left(B_n^T\Pi_n^{-2}B_n\right)^{-1}B_n^T\left(\boldsymbol{Z}_n - \tilde{\boldsymbol{Z}}_n\right)\right\|_2 = O_p\left(\sqrt{\frac{K_n^3 \log n}{n}}\right).$$

$$\tag{26}$$

Control of the second term in Equation (18): Notice that each coordinate of $\tilde{\boldsymbol{Z}}_n - \Pi_n^{-2}\tilde{\Pi}_n^2\tilde{\boldsymbol{Z}}_n$ is bounded, thus using similar arguments as that of Equation (21), we know that

$$\left\|\left(B_n^T\Pi_n^{-2}B_n\right)^{-1}B_n^T\left(\tilde{\boldsymbol{Z}}_n - \Pi_n^{-2}\tilde{\Pi}_n^2\tilde{\boldsymbol{Z}}_n\right)\right\|_2 = O_p\left(\sqrt{\frac{K_n^3\log n}{n}}\right). \tag{27}$$

Combining Equation (26) and (27) into (18), we know that

$$\|\hat{\boldsymbol{\alpha}} - \tilde{\boldsymbol{\alpha}}\|_2 = O_p\left(\sqrt{\frac{K_n^3\log n}{n}}\right),$$

and thus,

$$\|\hat{\epsilon}_n - \tilde{\epsilon}_n\|_{L^2} = O_p\left(\sqrt{\frac{K_n^2\log n}{n}}\right). \tag{28}$$

Combining the rate on bias and variance term, we get

$$\|\hat{\epsilon}_n - \check{\epsilon}_n\|_{L^2} = O_p\left(\rho_n + \sqrt{\frac{K_n^2\log n}{n}}\right) \overset{(a)}{=} O_p\left(K_n^{-2} + \frac{K_n\sqrt{\log n}}{\sqrt{n}}\right),$$

where (a) follows from assumption (iii), giving

$$\|\hat{\epsilon}_n - \check{\epsilon}_n\|_{L^2} = O_p\left(n^{-1/3}\sqrt{\log n}\right),$$

when taking $K_n \asymp n^{1/6}$. $\qquad\square$

### A.2.7 PROOF OF LEMMA 4

*Proof.* Suppose the SVD decomposition of $B_n = U\Lambda V^T$ where $U \in \mathbb{R}^{n\times n}$, $\Lambda \in \mathbb{R}^{n\times K_n}$, $V \in \mathbb{R}^{K_n\times K_n}$. From Lemma A.3 of Huang et al. (2004), we know that all diagonal elements of $(K_n/n)\Lambda^T\Lambda$ fall between some positive constants. Notice that the eigenvalues of $(K_n/n)B_n^T\Pi_n^{-2}B_n$ are the diagonal elements of $(K_n/n)\Lambda^T\Pi_n^{-2}\Lambda$. From the upper and lower boundedness of $\hat{\pi}_n$, we can get the desired conclusion. $\qquad\square$

### A.2.8 PROOF OF LEMMA 5

*Proof.* Write $\mathcal{F}_3 = \mathcal{F}_1 + \mathcal{F}_2, \mathcal{F}_4 = \mathcal{F}_1 - \mathcal{F}_2$. Notice that $\mathcal{F}_1\mathcal{F}_2 = \{f_1f_2 : f_1 \in \mathcal{F}_1, f_2 \in \mathcal{F}_2\} = \{\frac{1}{4}(f_1 + f_2)^2 - \frac{1}{4}(f_1 - f_2)^2 : f_1 \in \mathcal{F}_1, f_2 \in \mathcal{F}_2\} = \frac{1}{4}\mathcal{F}_3^2 - \frac{1}{4}\mathcal{F}_4^2$. Let $h : x \mapsto x^2$, from Theorem 12(4) of Bartlett & Mendelson (2002) we know that

$$\text{Rad}_n(\mathcal{F}_3\mathcal{F}_3) = \text{Rad}_n(h \circ \mathcal{F}_3) \leq 2\|\mathcal{F}_3\|_\infty\text{Rad}_n(\mathcal{F}_3).$$

Thus,

$$\begin{aligned}
&\text{Rad}_n(\mathcal{F}_1\mathcal{F}_2)\\
=&\text{Rad}_n(\frac{1}{4}\mathcal{F}_3^2 - \frac{1}{4}\mathcal{F}_4^2)\\
\leq&\text{Rad}_n(\frac{1}{4}\mathcal{F}_3^2) + \text{Rad}_n(-\frac{1}{4}\mathcal{F}_4^2)\\
\leq&\frac{1}{4}\text{Rad}_n(\mathcal{F}_3^2) + \frac{1}{4}\text{Rad}_n(\mathcal{F}_4^2)\\
\leq&\frac{1}{2}\|\mathcal{F}_3\|_\infty\text{Rad}_n(\mathcal{F}_3) + \frac{1}{2}\|\mathcal{F}_4\|_\infty\text{Rad}_n(\mathcal{F}_4)\\
\leq&\frac{1}{2}(\|\mathcal{F}_3\|_\infty + \|\mathcal{F}_4\|_\infty)(\text{Rad}_n(\mathcal{F}_3) + \text{Rad}_n(\mathcal{F}_4)).
\end{aligned}$$

$\qquad\square$

## A.3 Additional Results

This section formally states and proves the efficient influence function of a multidimensional vector, which is briefly mentioned in the main text. Suppose $\mathbf{\Gamma} = (\Gamma_1, \Gamma_2, \cdots, \Gamma_d)^T \in \mathbb{R}^d$ where

$$\Gamma_j = \int_{\mathcal{T}} \gamma^{(j)}(t; \mathbb{P}_T) \int_{\mathcal{X}} \int_{\mathcal{Y}} yp(y \mid \boldsymbol{x}, t)p(\boldsymbol{x})p(t)dydxdt.$$

Then we have the following theorem:

**Theorem 3.** *The efficient influence function for the d-dimensional vector $\mathbf{\Gamma}$ is*

$$\boldsymbol{\zeta}(Y, \boldsymbol{X}, T, \pi, \mu, \mathbf{\Gamma}) = (\zeta_1(Y, \boldsymbol{X}, T, \pi, \mu, \Gamma_1), \zeta_2(Y, \boldsymbol{X}, T, \pi, \mu, \Gamma_2), \cdots, \zeta_d(Y, \boldsymbol{X}, T, \pi, \mu, \Gamma_d))^T \in \mathbb{R}^d$$

*where $\zeta_j(Y, \boldsymbol{X}, T, \pi, \mu, \Gamma_j)$ is the efficient influence function for $\Gamma_j$, $j = 1, 2, \cdots, d$.*

*Proof.* Define $\mathbf{\Gamma}(\boldsymbol{\varepsilon}) := (\Gamma_1(\boldsymbol{\varepsilon}), \Gamma_2(\boldsymbol{\varepsilon}), \cdots, \Gamma_d(\boldsymbol{\varepsilon}))^T \in \mathbb{R}^d$ where for $i = 1, 2, \cdots, d$,

$$\Gamma_i(\boldsymbol{\varepsilon}) = \int_{\mathcal{T}} \gamma^{(i)}(t; \mathbb{P}_{T,\boldsymbol{\varepsilon}}) \int_{\mathcal{X}} \int_{\mathcal{Y}} yp(y \mid \boldsymbol{x}, t; \boldsymbol{\varepsilon})p(\boldsymbol{x}; \boldsymbol{\varepsilon})p(t; \boldsymbol{\varepsilon})dydxdt.$$

Define $\ell(Y, \boldsymbol{X}, T; \boldsymbol{\varepsilon}) = \log \mathbb{P}_{Y,\boldsymbol{X},T;\boldsymbol{\varepsilon}}$ where $\mathbb{P}_{Y,\boldsymbol{X},T;\boldsymbol{\varepsilon}}$ is a parametric submodel with parameter $\boldsymbol{\varepsilon} \in \mathbb{R}^d$ and $\mathbb{P}_{Y,\boldsymbol{X},T;\boldsymbol{0}} = \mathbb{P}_{Y,\boldsymbol{X},T}$. Then, the efficient influence function $\boldsymbol{\zeta}$ is defined as the unique function such that

$$\mathbb{E}\left[\boldsymbol{\zeta}\left(\frac{d\ell}{d\boldsymbol{\varepsilon}}\mid_{\boldsymbol{\varepsilon}=\boldsymbol{0}}\right)^T\right] = \frac{d\mathbf{\Gamma}(\boldsymbol{\varepsilon})}{d\boldsymbol{\varepsilon}}\mid_{\boldsymbol{\varepsilon}=\boldsymbol{0}}$$

i.e.,

$$\mathbb{E}\left[\zeta_i \frac{d\ell}{d\varepsilon_j}\mid_{\boldsymbol{\varepsilon}=\boldsymbol{0}}\right] = \frac{d\Gamma_i}{d\varepsilon_j}\mid_{\boldsymbol{\varepsilon}=\boldsymbol{0}}, \quad \forall i, j = 1, 2, \cdots, d$$

Notice that the efficient influence function $\zeta_i(Y, \boldsymbol{X}, T, \pi, \mu, \Gamma_i)$ for $\Gamma_i$ does not depend on $\boldsymbol{\varepsilon}$. Thus for each $i, j = 1, 2, \cdots, d$, the above equation can be proved using similar arguments as that in Section A.2.5. $\square$

## A.4 Experimental Details

### A.4.1 Network Structure

For all methods, we implement the conditional density estimator as a neural network with two hidden fully connected layers, each consisting of 50 hidden units using ReLU activation. Hidden feature $\boldsymbol{z}$ is defined as the latent representation extracted after the second ReLU activation. We set the number of grids $B = 10$. The estimation of $\pi(t \mid \boldsymbol{x})$ is computed as introduced in Section 3. Following Schwab et al. (2019), we use 5 blocks for Dragonnet and DRNet. Structure of prediction head for each block is the same as the prediction head $\mu$ for VCNet, except that Dragonnet and DRNet do not use treatment-dependent weights. In VCNet, the prediction head for $\mu(t, \boldsymbol{x})$ is a neural network with two hidden fully connected layers stacking over the hidden feature $\boldsymbol{z}$. Each hidden layer consists of 50 hidden units with ReLU activation. We use B-spline with degree two and two knots placed at $\{1/3, 2/3\}$ (altogether 5 basis). In this way, all methods have the same complexity, i.e., the number of parameters. We also tried different structures and found the relative performance of different methods to be similar. Thus, all reported results below are based on this structure. All networks are trained for 800 epochs.

### A.4.2 Parameter Setting

For each dataset we tune parameters based on 20 runs. In each run we simulate data, randomly split into training and testing, and use AMSE on testing data for evaluation. We tune the following parameters. For all methods: network learning rate lr $\in \{0.05, 0.005, 0.001, 0.0005, 0.0001\}$ and $\alpha \in \{1, 0.5\}$. For TR: learning rate for $\epsilon(t)$: $\text{lr}_\epsilon \in \{0.001, 0.0001\}$, $\beta \in \{20, 10, 5\} \times n^{-1/2}$. We found that performance is not sensitive to $\alpha$. In estimator of $\epsilon(t)$, we use B-spline with degree 2 and tune the number of knots across $\{5, 10, 20\}$ (all equally spaced at $[0, 1]$). For TMLE and doubly robust estimator: we tune parameters of B-spline in the same way as in TR version. During tuning, all networks are trained for 800 epochs.

### A.4.3 STATISTICAL BASELINES

We implement Causal forest (Wager & Athey, 2018) using R package 'grf' (Tibshirani et al., 2018), BART using R package 'bartMachine' (Kapelner et al., 2016), and GPS using R package 'causaldrf' (Galagate et al., 2015). We tune the paramters of each method on each dataset using 20 separate tuning sets, including the number of trees for BART, the number of trees and minimum node size for causal forest, and the number of knots for GPS. The other hyper-parameters are set to the default value of the R packages.

### A.4.4 DATASET

**Synthetic Dataset**   We generate data as follows: $x_j \overset{\text{i.i.d.}}{\sim} \text{Unif}[0,1]$, where $x_j$ is the $j$-th dimension of $\boldsymbol{x} \in \mathbb{R}^6$, and

$$\widetilde{t} \mid \boldsymbol{x} = \frac{10 \sin(\max(x_1, x_2, x_3)) + \max(x_3, x_4, x_5)^3}{1 + (x_1 + x_5)^2} + \sin(0.5 x_3)(1 + \exp(x_4 - 0.5 x_3))$$
$$+ x_3^2 + 2\sin(x_4) + 2x_5 - 6.5 + \mathcal{N}(0, 0.25),$$
$$y \mid \boldsymbol{x}, t = \cos(2\pi(t - 0.5)) \left( t^2 + \frac{4 \max(x_1, x_6)^3}{1 + 2x_3^2} \sin(x_4) \right) + \mathcal{N}(0, 0.25),$$

where $t = (1 + \exp(-\tilde{t}))^{-1}$. Notice that $\pi(t \mid \boldsymbol{x})$ only depends on $x_1, x_2, x_3, x_4, x_5$ while $Q(t, \boldsymbol{x})$ only depends on $x_1, x_3, x_4, x_6$. As discussed in Shi et al. (2019), this allows us to observe the improvement using VCNet when noise covariates exist. Results are reported in Table 1.

**IHDP**   The original semi-synthetic IHDP dataset from Hill (2011) contains binary treatments with 747 observations on 25 covariates. To allow comparison on continuous treatments, we randomly generate treatment and response using:

$$\widetilde{t} \mid \boldsymbol{x} = \frac{2x_1}{(1 + x_2)} + \frac{2 \max(x_3, x_5, x_6)}{0.2 + \min(x_3, x_5, x_6)} + 2 \tanh\left( 5 \frac{\sum_{i \in S_{\text{dis},2}} (x_i - c_2)}{|S_{\text{dis},2}|} \right) - 4 + \mathcal{N}(0, 0.25),$$
$$y \mid \boldsymbol{x}, t = \frac{\sin(3\pi t)}{1.2 - t} \left( \tanh\left( 5 \frac{\sum_{i \in S_{\text{dis},1}} (x_i - c_1)}{|S_{\text{dis},1}|} \right) + \frac{\exp(0.2(x_1 - x_6))}{0.5 + 5 \min(x_2, x_3, x_5)} \right) + \mathcal{N}(0, 0.25),$$

where $t = (1 + \exp(-\tilde{t}))^{-1}$, $S_{\text{con}} = \{1, 2, 3, 5, 6\}$ is the index set of continuous features, $S_{\text{dis},1} = \{4, 7, 8, 9, 10, 11, 12, 13, 14, 15\}$, $S_{\text{dis},2} = \{16, 17, 18, 19, 20, 21, 22, 23, 24, 25\}$ and $S_{\text{dis},1} \cup S_{\text{dis},2} = [25] - S_{\text{con}}$. Here $c_1 = \mathbb{E}\frac{\sum_{i \in S_{\text{dis},1}} x_i}{|S_{\text{dis},1}|}$, $c_2 = \mathbb{E}\frac{\sum_{i \in S_{\text{dis},2}} x_i}{|S_{\text{dis},2}|}$. Notice that all continuous features are useful for $\pi(t \mid \boldsymbol{x})$ and $Q(t, \boldsymbol{x})$ but only $S_{\text{dis},1}$ is useful for $Q$ and only $S_{\text{dis},2}$ is useful for $\pi$. Following Hill (2011), covariates are standardized with mean 0 and standard deviation 1 and the generated treatments are normalized to lie between $[0, 1]$. Results are summarized in Table 1.

**News**   The News dataset consists of 3000 randomly sampled news items from the NY Times corpus (Newman, 2008), which was originally introduced as a benchmark in the binary treatment setting (Johansson et al., 2016). We generate the treatment and outcome in a similar way as Bica et al. (2020). We first generate $\boldsymbol{v}_1'$, $\boldsymbol{v}_2'$ and $\boldsymbol{v}_3'$ from $\mathcal{N}(\boldsymbol{0}, \boldsymbol{1})$ and then set $\boldsymbol{v}_i = \boldsymbol{v}_i' / \|\boldsymbol{v}_i'\|_2$ for $i = \{1, 2, 3\}$. Given $\boldsymbol{x}$, we generate $t$ from $\text{Beta}\left( 2, \left| \frac{\boldsymbol{v}_3^\top \boldsymbol{x}}{2\boldsymbol{v}_2^\top \boldsymbol{x}} \right| \right)$. And we generate the outcome by

$$y' \mid \boldsymbol{x}, t = \exp\left( \frac{\boldsymbol{v}_2^\top \boldsymbol{x}}{\boldsymbol{v}_3^\top \boldsymbol{x}} - 0.3 \right),$$
$$y \mid \boldsymbol{x}, t = 2 \left( \max(-2, \min(2, y')) + 20 \boldsymbol{v}_1^\top \boldsymbol{x} \right) * \left( 4(t - 0.5)^2 * \sin\left( \frac{\pi}{2} t \right) \right) + \mathcal{N}(0, 0.5).$$

