# OpenReview forum: "VCNet and Functional Targeted Regularization For Learning Causal Effects of Continuous Treatments"
_ICLR.cc/2021/Conference — ICLR 2021 Oral_

### Official Review · AnonReviewer3 · 2020-10-28
**This paper focuses on the problem of estimating average dose-response curve from observational data. The paper proposes 1) varying coefficient neural network, a network designed for continuous treatment; 2) functional targeted regularization, a doubly robust estimator of the whole ADRF curve.**

**Rating:** 9
**Confidence:** 4

**Review:**

This problem is well-motivated --- estimating dose-response is a challenging and practically important problem.
The paper is extremely well written.  It explained complex (poor written) ideas in the semiparametric literature clearly.
The comparison against existing works is clear.
The theory, as far as I can tell, is sound. It improves the existing results in targeted regularization and can be adapted to analyze one step TMLE.
The experiment shows that the model outperforms existing benchmarks on the task it set out to do.

My main suggestion to improve the paper is to use some datasets that actually have continuous treatment in evaluating the method.  In particular, I would be interested in seeing an application of the methods on real-world datasets. That being said, while it will improve the paper, it's probably asking too much for an 8-page conference submission. I believe the paper at its current state is sufficient for acceptance.

I want to thank the authors for writing such an elegant paper. I really enjoyed reading it.

---

> ### Author Response · Authors · 2020-11-18
> **Response to Reviewer 3**
>
> We thank the reviewer for the encouraging and constructive comments. We tried searching for a suitable dataset with continuous treatment when doing the project. But the problem is that in the real world, it is hard (probably impossible) to obtain the true ADRF (as we can’t observe the counterfactual result in observational study) and thus we fail to find a real world dataset. Thus we follow the same experiment setting as previous work [1,2] in which we consider a semi-synthetic dataset.
>
> **Reference**
>
> [1] Claudia Shi, David Blei, and Victor Veitch. Adapting neural networks for the estimation of treatment effects. In Advances in Neural Information Processing Systems, pp. 2503–2513, 2019.
>
> [2] Patrick Schwab, Lorenz Linhardt, Stefan Bauer, Joachim M Buhmann, and Walter Karlen. Learning counter- factual representations for estimating individual dose-response curves. arXiv preprint arXiv:1902.00981, 2019.

---

### Official Review · AnonReviewer2 · 2020-10-29
**Varying Coefficient Neural Network with Functional Targeted Regularization for Estimating Continuous Treatment Effects**

**Rating:** 6
**Confidence:** 5

**Review:**

This paper is to develop a varying coefficient neural network to estimate average dose-response curve (ADRF).  Although this paper has several interesting results, the paper is full of many typos and small errors.  The current paper needs substantial improvement. Here are some detailed comments.

1. The introduction section is not well written since the logic does not flow very smooth.
2. The motivation for （1）can be improved. You miss period in (1).
3. In Theorem 2, please not use (1)-(6) to itemize the assumptions, since you use them to denote the equations.
4. In the proof of all theorems, there are some obvious mistakes inside.

---

> ### Author Response · Authors · 2020-11-18
> **Response to Reviewer 2**
>
> We thank the reviewer for the constructive feedback. We uploaded a revision of the manuscript, in which we make efforts to address your concerns:
> * We reorganize the introduction to make the logic flow more smoothly.
> * We add a sentence to explain the motivation of using loss (1) in section 3.3 of the revision. Notice that this loss is also used in [1].
> * We change the index of assumptions in Theorem 2.
> * We also checked the proofs thoroughly and corrected several typos and improved the presentation of the proofs. We would appreciate it if you could check and point out any remaining bugs, if any.
>
> **Reference**
>
> [1] Claudia Shi, David Blei, and Victor Veitch. Adapting neural networks for the estimation of treatment effects. In Advances in Neural Information Processing Systems, pp. 2503–2513, 2019.

---

### Official Review · AnonReviewer4 · 2020-10-29
**This paper aims at estimating average dose-response curve (ADRF) and propose to design a new varying coefficient neural network and a doubly robust estimator for ADRF.**

**Rating:** 5
**Confidence:** 4

**Review:**

Solid theoretical results are provided to confirm the doubly robustness of the treatment estimator and outcome estimator. This paper is well-written, however, I still have some concerns about the contributions.
1.	The author claimed that confounder is one challenge for treatment effect estimation. Some existing deep learning methods resort to balanced learning or reweighting. For the continuous treatment, how the proposed method addresses the confounder factors?
2.	Another big concern is the varing coefficient neural network. Why the VCnet designed to be dependent of treatment information? The dependence is not theoretically discussed in this paper.
3.	The experimental results are not convincing for me. Only two baselines are compared with the proposed method. Also the sufficient theoretical discussion is given which however is mainly about the doubly robust estimators. But the design of VCnet needs the numerical results to confirm its effectiveness for ADRF.

---

> ### Author Response · Authors · 2020-11-18
> **Response to Reviewer 4 [1/2]**
>
> We thank you for your constructive comments! Below please find our response. We hope that our responses to your concerns will increase your enthusiasm for our paper. If you believe we haven’t addressed all your concerns, we look forward to further discussions with you.
>
> **Q1**:
> The author claimed that confounder is one challenge for treatment effect estimation. Some existing deep learning methods resort to balanced learning or reweighting. For the continuous treatment, how the proposed method addresses the confounder factors?
>
> **A1**:
> First, we assume there are no unobserved confounders. Under assumption 1, our method is based on the following expression of ADRF: $\psi(t)=\mathbb{E}_{\boldsymbol{X}}[\mu(\boldsymbol{X},t)]$. Notice that the above expectation is taken with respect to the marginal distribution of $X$.
> This implies that a naive method to estimate ADRF is to obtain an estimator $\hat\mu$ for $\mu(\boldsymbol{x},t)=E(Y|\boldsymbol{X}=\boldsymbol{x},T=t)$ and then use
>
> $\hat\psi(t)=1/n\sum_{i=1}^n\hat\mu(\boldsymbol{x}_i,t)$
>
> as the estimator for $\psi(t)$. Here estimating $\mu=E(Y|\boldsymbol{X},T)$ does not require reweighting, which is also a common practice in previous literature, see, for example [1, 2]. And since we are directly using the empirical marginal distribution of $X$ (to approximate the population marginal distribution of $X$), reweighting is also not needed. We refer Reviewer 1 to Section 2 and the first paragraph of Section 3 of the main text for more details.
>
> **Q2**:
> Another big concern is the varying coefficient neural network. Why the VCNet designed to be dependent of treatment information? The dependence is not theoretically discussed in this paper.
>
> **A2**:
> We are not sure whether we correctly understand your concern on ‘Why the VCNet designed to be dependent on treatment information’. We believe you inquire why we make the parameters of the network depend on treatment.
> The treatment is an input to the network but it is put in a special position. The reason we make it special is to prevent it from being lost in high dimensional latent representations as discussed in [3], which also motivates previous network structures such as [1] and [2].  Please note that, as discussed in the last paragraph of Section 3.1 in the main text, the previous structure can also be viewed as VCNet in which the parameters of the network also depend on the treatment. The difference with the proposed VCNet is that they use a suboptimal choice of spline basis, which breaks the continuity structure of ADRF. So if previous lines of work on network structure make sense, we believe the proposed VCNet should also do.
>
> In terms of a theoretical justification, as discussed in paragraph 3 of introduction and Fig 1 in the original version of the submission, we believe one theoretical advantage of VCNet is that it takes the continuity of ADRF into modeling and thus produces better estimators in practice. In terms of more comprehensive theoretical analysis on why architecture A is better than B, as far as we know, there are no mathematical tools to theoretically study it at the current stage. For example, there are various building blocks proposed in the CV and NLP area but current theoretical tools could not explain why a certain building block is better than another.

---

> ### Author Response · Authors · 2020-11-18
> **Response to Reviewer 4 [2/2]**
>
> **Q3**:
> The experimental results are not convincing for me. Only two baselines are compared with the proposed method. Also the sufficient theoretical discussion is given which however is mainly about the doubly robust estimators. But the design of VCnet needs the numerical results to confirm its effectiveness for ADRF.
>
> **A3**:
> We respectfully disagree with the noted lack of experimental results.
>
> First of all, there are more than two baselines. We improve the estimation of ADRF in two-fold: better architecture and functional targeted regularization. And thus our baselines are previous architectures and different methods for estimation (naive, doubly robust, TMLE). Table 1 can be read both column-wise and row-wise and thus there are altogether 3*4=12 methods implemented for each dataset. In Table 1, each row compares the performance of different versions of estimators under the same architecture, and each column compares the performance of different architectures under the same version of estimator. Besides, the compared architectures are sota for treatment effect estimation.
>
> Secondly, our experiment exactly gives numerical results to confirm the effectiveness of VCNet for estimating ADRF. Every column of Table 1 compares the performance of VCNet against the old architecture in estimating ADRF while holding everything else the same, i.e., the only thing that changes within each column is the network structure. Thus, by comparing within each column of Table 1, we see that VCNet improves over the old network architectures in all three datasets. Besides, our design of experiment also shows that VCNet improves previous architectures for all versions of estimators (by looking at all columns in the table). We believe the current result is sufficient to show the effectiveness of VCNet.
>
>
> **Reference**
>
> [1] Claudia Shi, David Blei, and Victor Veitch. Adapting neural networks for the estimation of treatment effects. In Advances in Neural Information Processing Systems, pp. 2503–2513, 2019.
>
> [2] Patrick Schwab, Lorenz Linhardt, Stefan Bauer, Joachim M Buhmann, and Walter Karlen. Learning counter- factual representations for estimating individual dose-response curves. arXiv preprint arXiv:1902.00981, 2019.
>
> [3] Shalit, Uri, Johansson, Fredrik D, and Sontag, David. Estimating individual treatment effect: generalization bounds and algorithms. In Proceedings of the 34th International Conference on Machine Learning-Volume 70, pp. 3076–3085. JMLR. org, 2017.

---

### Author Response · Authors · 2020-11-21
**Summary of revision**

We uploaded revisions of the submission in which

* We reorganize the introduction to make the logic flow more smoothly.
* We add a sentence to explain the motivation of using loss (1) in section 3.3 of the revision.
* We change the index of assumptions in Theorem 2.
* We also checked the proofs thoroughly and corrected several typos and improved the presentation of the proofs.
* We make a few editing on abstract to improve the presentation.

---

### Decision · Program_Chairs · 2021-01-07
**Final Decision**

**Decision:**

Accept (Oral)

**Comment:**

The paper designs a new way (in some sense a new perspective) on how neural networks can be used to model intervention variables when the goal is to estimate ADRF.  Basically, the idea is to emphasize the importance of the intervention variable by ensuring that it appears not just in every layer but also in every neural of a neural network.

Reviewers mostly agree that this is a good paper with varying degrees, although there are some criticisms on e.g., assuming away the confounders.  However, I believe the authors address the criticisms of R4 satisfactorily.

Overall I find the idea new and interesting and the experimental results strong, hence I happily recommend accepting the paper.

I do have a few quips myself and some comments that may help the authors to further improve the paper.

1. Re: the design that models each parameter as a spline.
This is equivalent to introducing additional parameters (coefficients for spline basis) and adding a fixed linear layer (spline basis themselves) to every layer of the neural networks. t is taken as an input in all layers thus it makes sure that the model prioritizes on learning the impact of t.

2. If you use a B-spline basis (that comes with kernels of bounded support), then the proposed method is very similar to stratifying the data according to different bins of t, and then fitting a separate model for each t. The only difference is that the different bins are now smooth kernels and they overlap somewhat. As a side note,  the authors should clearly write out how they are choosing the knots to specify the basis functions. Otherwise the paper will not be reproducible.

3. I am not sure how this method would compare to naive (non-deep) baselines. Maybe this was considered in a prior work? If not, then I tend to side with Reviewer 4 that the evaluations are mostly ablation studies and they are not really comparing to representative work in this domain. Given that there is a large body of work on this before deep learning takes over, it is important to somehow compare with the right baselines.

---

> ### Author Response · Authors · 2021-03-14
> **Camera Ready Revisions**
>
> We thank the program chair and all reviewers very much for their suggestions and comments! In the uploaded camera ready version, we have addressed the comments raised by reviewers and program chair. Specifically, we do the following updates:
>
> 1. In Section 6, we add more details on the experimental setups and in particular, we write explicitly on how we choose the knots. The other details on parameter settings / network structure are included in the Appendix.
>
> 2. We added a comparison to non-deep baselines and the result is now included in Section 6. Parameters setting of non-deep baselines are also added in the Appendix.
>
> 3. We added link to an open source repository that contains code and data of our experiments.
>
> 4. Typos in the main text as well as Appendix are now fixed.
>
> Thanks again to the reviewers and program chair for their suggestions!